# Star Power: Early life stages of an endangered sea star are robust to current and near-future warming

Jason Hodin[1]*, Fleur P. Anteau[1], Brook F. Ashcraft[1], Michael Brito[1,2], Fiona Curliss[1], Augustin R. Kalytiak-Davis[3], James S. Peng[4], Chloe J. Schwab[1], Vannessa V. Valdez[1], Willem Weertman[1,5]

1 Friday Harbor Laboratories, University of Washington, Friday Harbor, Washington, United States of America, 2 Exploratorium, San Francisco, California, United States of America, 3 Department of Integrative Biology, Oregon State University, Corvallis, Oregon, United States of America, 4 Department of Biostatistics, University of Washington, Seattle, Washington, United States of America, 5 Department of Psychology, University of Washington, Seattle, Washington, United States of America

* larvador@uw.edu

## Abstract

The sunflower star, *Pycnopodia helianthoides*, was a top benthic predator throughout its former range from Alaska to northern Mexico, until its populations were devastated starting in 2013 by a disease known as seastar wasting. The subsequent absence of sunflower stars from northern California waters was coincident with a dramatic ecological phase shift from healthy bull kelp forests (*Nereocystis luetkeana*) to barrens formed by purple sea urchins (*Strongylocentrotus purpuratus*), a prey of sunflower stars. Modeling suggests that restoration and resilience of kelp forests can be enhanced by the return of sunflower stars. Towards this end we run a conservation breeding program for sunflower stars in the Salish Sea of Washington, where sunflower stars have persisted in much reduced numbers. We here report on a variety of investigations into the temperature tolerance of sunflower stars, focusing on their poorly studied early life stages from their planktonic embryos and larvae, through metamorphosis and settlement as they transition to the benthos, and then for eight months of juvenile growth. Our results indicate that the optimum temperature for early life stage sunflower stars is more than 4°C higher than ambient temperatures in the Salish Sea, and that the juveniles demonstrate enhanced performance to a simulated marine heat wave. These results suggest that Salish Sea-derived sunflower stars would be robust to current and even near-future predicted temperatures in the south of their former range.

**Data availability statement:** The data underlying the results presented in the study are publicly accessible at: https://doi.org/10.5061/dryad.msbcc2g9s

**Funding:** JH – The Nature Conservancy of California (TNC-CA; no funding number), participated in conversations about some aspects of study design; TNC-CA did not participate in any data collection, analyses or manuscript preparation. JH – California Sea Grant (CSG) and the Ocean Protection Council (CSG award number R/HCE-15), did not participate in any aspects of study design, data collection, data analysis or manuscript preparation.

**Competing interests:** The authors have declared that no competing interests exist.

## Introduction

Ongoing and future climate change threatens the resilience of marine ecosystems and also obscures population forecasts in taxa of interest, such as keystone species, primary producers and habitat engineers [1,2]. Focusing on individual species of concern in a laboratory context can be edifying yet can also pose challenges for interpretation. One can design lab experiments to model future climate scenarios [3], investigating some measure of relative fitness (e.g., growth rates, survival, reproductive output), under a variety of conditions (e.g., predicted mean or extreme temperature conditions, multiple stressors). However, such experiments often focus on only one life history stage, which can paint an incomplete picture of the effect across the complex life cycles of most multicellular organisms [4,5]. Furthermore, it is difficult to determine whether evolutionary change in a particular species could track the rate and scope of future climate change [6]. As a result of these uncertainties, it can be challenging to predict how a given taxon will fare under various future climate scenarios [7].

One possible approach to overcome at least a subset of these challenges is to examine organisms' performance across different stages of ontogeny [4,5]. For example, if a study on adults suggests a sensitivity to high temperatures that is not seen in the larvae of that species, this could indicate a capacity for evolutionary change under future selection. Since the genome of this hypothesized organism has the cellular capability to perform well under elevated temperatures, then adults could potentially gain the capacity to likewise cope with high temperatures under strong positive selection.

Studies of climate resilience across ontogeny are also informative because most multicellular organisms occupy different habitats at different life phases, with selection occurring at all stages [8]. As such, a complete understanding of the environmental resilience of a given taxon is impossible without considering the entire life cycle [4]. The importance of considering the full life cycle is especially acute in the ocean, where complex life cycles are the norm. Nevertheless, the majority of published work focuses on early embryos and reproductive adults, with far fewer studies examining what can be many years of ontogeny in between [5]. Here we adopt this logic in the context of investigating the performance of early life stages of an endangered species [9], the sunflower star *Pycnopodia helianthoides* (Brandt, 1835), under a broad range of environmental temperatures.

Sunflower stars historically ranged from southeast Alaska to Baja California, and were a top benthic predator throughout this range in a variety of habitats including shallow rocky intertidal, subtidal kelp forests and eelgrass meadows, and deeper waters [10]. Starting in 2013, sunflower star populations were drastically reduced by an unprecedented outbreak of seastar wasting (SSW) that impacted dozens of sea star (i.e., 'asteroid') species along thousands of kilometers of northeast Pacific coastline [11–13]. Among all exposed asteroids, sunflower stars were impacted the most severely, and now have the unwelcome distinction of being the first sea star species ever 'red-listed' by the International Union for the Conservation of Nature (IUCN) as critically endangered [9]. More recently, the National Oceanographic and Atmospheric

Administration (NOAA) has proposed listing sunflower stars as 'threatened' under the Endangered Species Act [14], another first for any asteroid.

Sunflower star declines occurred across their range, but losses were more severe in the south: the stars were all but extirpated in Baja México and throughout the state of California, and faced near elimination off the coast of Oregon [15,16]. Despite some recent unpublished reports to the contrary, there remains little documented evidence for increasing numbers except perhaps in some locations towards the north of their former range [17].

Predators are well known to structure ecosystems through top-down control [18,19]. As the sunflower star is a top benthic predator, it is not surprising that the impacts of its near total disappearance in the southern half of its historic range has had ecosystem level consequences. A persistent regional warm water event known as 'the blob' [20] was coincident with the onset of the SSW pandemic in 2013. The combined result was an extreme ecological phase shift [21] involving a precipitous decline in healthy bull kelp (*Nereocystis luetkeana*) forests and an explosion in populations of kelp-eating and barren-forming purple urchins, *Strongylocentrotus purpuratus* [22]. Other than sea otters (*Enhydra lutris*), which remain absent through much of the northeast Pacific [23], sunflower stars are the only known major predator of purple urchin adults north of central California [24]. Indeed, recent experimental and modeling results indicate that predation by adult sunflower stars can keep purple urchin numbers in check, thus protecting kelp [25,26].

Furthermore, laboratory studies [27,28] indicate that sunflower star juveniles are juvenile urchin predators starting at the very earliest juvenile (i.e., post-settlement) stages of both species. Our unpublished data indicate that sunflower star juveniles can consume more than 10x as many juvenile urchins per day compared to sunflower star adults consuming adult urchins. In this sense, considering interactions between sunflower stars and urchins throughout their respective benthic life stages could bolster our understanding of the importance of sunflower stars in maintaining kelp forest health, and again speaks to the importance of considering the full life cycle.

Such findings suggest that restoration of sunflower stars throughout their historical range will offer future resilience to kelp forests. But what of the impacts of climate change? Can we expect this endangered species to recover and thrive in a changing ocean?

Here we report on our initial attempt to address these questions, looking specifically at the performance of the under-studied early life stages of sunflower stars: pre-feeding embryos, planktotrophic larvae, the metamorphic and settlement stages as they transform into predatory juveniles, and through eight months of post-settlement juvenile ontogeny. Our results indicate that all of these early life stages of sunflower stars are quite robust over a broad range of experimental temperatures (10–20°C), including mean and extreme ocean temperatures predicted under a range of future warming scenarios for the northeast Pacific. Interestingly, our experiments surrounding the metamorphosis phase revealed a novel trade-off between larval and juvenile structures at higher temperatures. This shift in investment is reminiscent of the well-studied effect of low food on planktotrophic larvae [29]. In sum, our results bode well for the future resilience of sunflower stars to at least one major aspect of the changing ocean.

## Results

### Experiments overview

We report on six separate experiments (singular "Exp"; plural "Exps") conducted in 2021–2023: Embryo and Larval Growth Exp (2021), Pre-Settlement Temperature Shift Exp (2021), Settlement Exp (2023), Juvenile Growth Exp 1 (2022), Juvenile Growth Exp 2 (2023), and Juvenile Performance Exp (2023). Key parameters, abbreviations and internal Figure and Table references for these six experiments are outlined in Table 1, and the particular protocols for each experiment detailed in the Methods.

### Exp 1–Larv 2021

**Embryos and larvae grow optimally at high temperature, with an intriguing switch as larvae approach the settlement stage.** We conducted a series of short-term temperature exposures in replicate jars at three different

**Table 1. Summary of the six experiments described herein.**

| Experiment ("Exp") | Year | Experiment abbreviation | Factors assessed | Fig/ Table reference |
|---|---|---|---|---|
| Embryo and Larval Growth Exp (embryo: 10–18°C; larvae: 12–20°C) | 2021 | *Exp 1– Larv 2021* | embryo & larval morpho-metrics, incipient juvenile characters, cloning, settlement | Figs 1-4,10, Tables 2–3, S1–S5 Figs, S1–S3 Tables |
| Pre-Settlement Tem- perature Shift Exp (12,16°C) | 2021 | *Exp 2– Temp Shift Settle 2021* | settlement | Fig 5 |
| Settlement Exp (11,14,17°C) | 2023 | *Exp 3–Settlement 2023* | size and morphology at settlement | Figs 3B, 6, S4 Table |
| Juvenile Growth Exp 1 (11,16°C) | 2022 | *Exp 4– Juv Growth 2022* | post-settlement juvenile growth and survival | Figs 6B, 7A |
| Juvenile Growth Exp 2 (11,17°C) | 2023 | *Exp 5– Juv Growth 2023* | post-settlement juvenile growth and survival | Figs 6B, 7B |
| Juvenile Performance Exp (12,14.5,17°C) | 2023 | *Exp 6– Flipping 2023* | Juvenile righting behavior, 7–8 months post-settlement | Figs 8, 9, S6 Fig |

Note the numbered experiment abbreviations (*in Italics*) used throughout for convenience.

developmental stages in sunflower stars –embryogenesis, bipinnaria (mid larva), brachiolaria (late larva)– and conducted a variety of morphometric measurements and scoring criteria at each stage (Fig 1, Table 2, S1 Table), analyzing them by principal component (PC) analysis.

At the embryo stage, we exposed embryos to 10, 12, 14, 16 or 18°C for 5 days starting 2 days post fertilization (2 dpf). As ambient temperatures in the Salish Sea are between 10 and 12 degrees during the time these embryos are in the plankton, we predicted that larvae would grow optimally at these lower temperatures. That is not what we observed. The optimal temperature for the composite PC1 variable in the embryo stage was 18.0°C (95% C.I.: 16.7–18.0°C; Fig 2, Table 3).

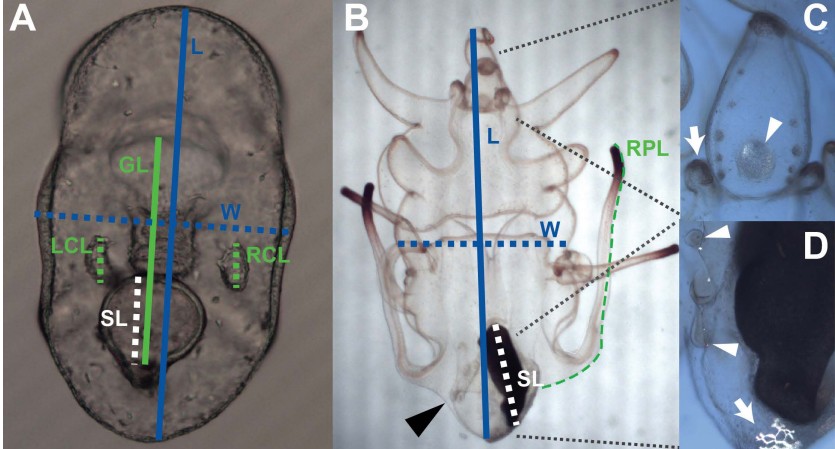

**Fig 1. Morphometric measurements and other structures scored in *Exp 1*. (A)** Late stage embryo features: length (L; *blue solid line;* 0.6 mm*), w*idth (W, *blue dotted line*), stomach length (SL; *white dotted line*), gut length (GL; *green solid line*) and right and left coelom lengths (RCL, LCL; *green dotted lines*). Stomach width not shown, but perpendicular to and bisecting the mid point of SL line. **(B)** Brachiolaria stage larva (abbreviations and colors as in A unless noted) indicating length (2.8 mm), width, stomach length and right posterodorsal arm length (RPL; *green dotted line*). Stomach width (see above) and LPL not shown. **(C)** Closeup of anterior region of brachiolaria larva under cross-polarized illumination showing birefringent, mature attachment disk (*arrowhead*) and brachiolar arm buds (*arrow*). The grey circles to the left and right of the attachment disk are the 'side pads'. **(D)** Closeup of anterior region of brachiolaria larva under cross-polarized illumination showing a 'snowflake' stage skeletal plate (*arrow*) and radial canals (*arrowheads*). Note the skeletal spicules adjacent to the radial canals. See S1 Table for further descriptions.

**Table 2. Details of experimental design for *Exp 1–Larv 2021*.**

| developmental stage analyzed | exposure temperatures (°C) | replicates (jars) per temp | embryos/ larvae per jar | mls per jar | hpf or dpf exposures started | hpf or dpf exposures ended | morphometrics at __ hpf or dpf |
|---|---|---|---|---|---|---|---|
| embryo-genesis | 10, 12, 14, 16, 18 | 3 | 200 | 600 | 48 hpf (2 dpf) | 164 hpf (~7 dpf) | 48, 66, 92, 118, 142 and 164 hpf |
| bipinnaria | 12, 14, 16, 18, 20 | 3 | 60 | 600 | 26 dpf | 34 dpf | 26, 30, 34 dpf |
| brachiolaria | 12, 14, 16, 18, 20 | 3 | 80 | 600 | 51 dpf | 64 dpf | 51, 64 dpf |

hpf–hours post fertilization; dpf–days post fertilization.

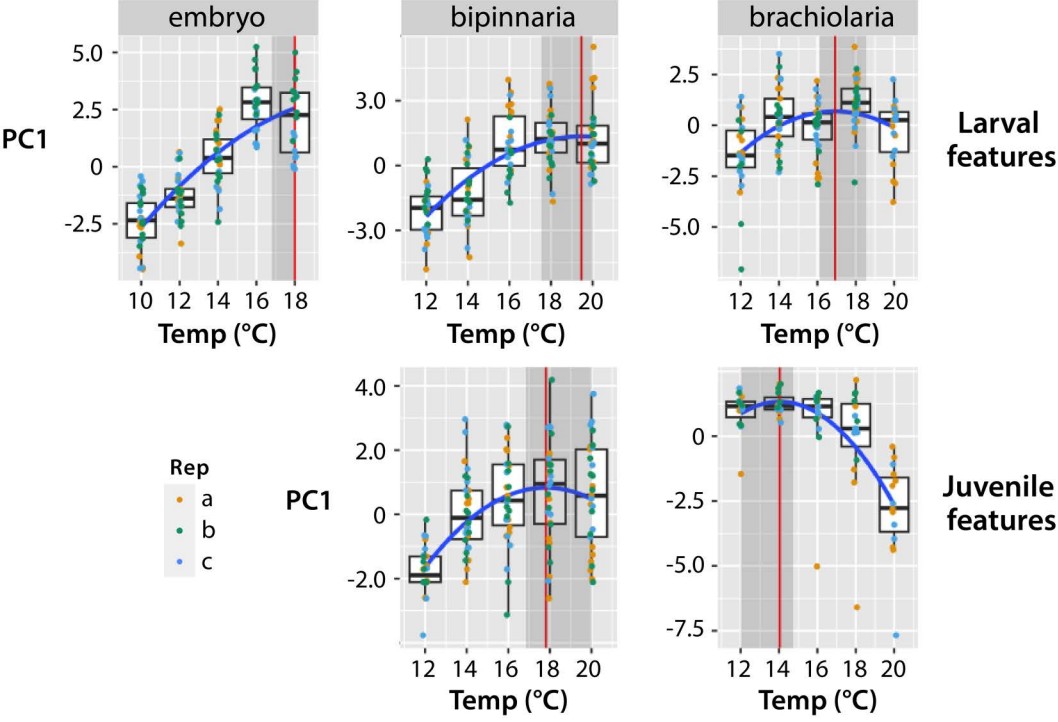

**Fig 2. Growth and development of larval and juvenile characters in *Exp 1*, represented by PC1 composite variables in each stage across different temperature exposures.** *Upper three graphs* – larval characters; *lower two graphs* – juvenile characters. Individual points represent individual larvae, with different colors for each replicate. *Blue line* shows quadratic fit. *Red vertical lines* and *shaded regions* show the estimate and 95% confidence interval for the optimal temperature in each graph.

Additionally, the estimated temperature optimum for each individual character was approximately 18°C, the highest temperature condition tested (S1 Fig).

Note that at the embryo stage we observed a clear jar effect, where one jar of embryos at one temperature (18°C replicate A) was clearly developing abnormally relative to all other jars, irrespective of temperature. The embryos in this jar did not undergo gastrulation normally, and contained apparently necrotic cells. Statistical analysis on the morphometric measurements confirmed that this replicate was an outlier. We feel confident that this was a bonafide jar effect, perhaps caused by some teratogenic compound residue in this particular jar. For the data presented in Fig 2, Table 3 and S1 Fig, we dropped this replicate from the analysis.

**Table 3.** *Exp 1–Larv 2021* optimum *temperatures.*

| Stage | Optimal temperature in °C (95% C.I.) |
|---|---|
| Embryo | |
| Larval characters | 18.0 (16.7-18.0) |
| Bipinnaria | |
| Larval characters | 19.5 (17.6-20.0) |
| Juvenile characters | 17.8 (16.8-20.0) |
| Brachiolaria | |
| Larval characters | 16.9 (16.1-18.8) |
| Juvenile characters | 14.0 (12.0-14.8) |

Bootstrapped 95% confidence intervals for optimal temperature (in °C) for larval and juvenile characters, using the PC1 composite variable at each stage, employing the 2.5% and 97.5% quantiles of the bootstrap results.

At the bipinnaria stage, we shifted the temperature exposure series up by 2°C based on the unexpected high temperature optimum for embryos. Therefore at this stage we exposed larvae to 12, 14, 16, 18 or 20°C for 8 days until 34 dpf. Here we also included some additional morphological features in the analysis which had not yet appeared during the embryonic stages, including some juvenile skeletal features that had begun to develop at the bipinnaria stage (Fig 1, S1 Table). The optimal temperature for growth and development of all characters largely matched what we observed for the embryo stage characters. Optimal temperature was 19.5°C (95% CI: 17.6–20.0°C) for larval features and 17.8°C (95% CI: 16.8–20.0°C) for juvenile features (Fig 2; Table 3). S2 and S3 Figs present data for the individual larval and juvenile characters, respectively.

At the brachiolaria stage, we allowed the larvae to develop at the target temperatures for 13 days until 64 dpf. During that time, many of the larvae underwent larval cloning, a common phenomenon in echinoderms [30], and in particular sea stars [31]. The result is fission along various larval axes, resulting in a wide range of larval sizes and morphologies. One result of this extensive cloning is that morphometric analyses become challenging at best. To address this challenge, we classified each larva in each of the 15 replicate jars on 64 dpf into either fully developed larvae or one of three classes of clones (see Methods for definitions). We provide the raw data and summary statistics for these four classes in S2 Table. To examine whether we detected a temperature effect on cloning rates, we combined the three cloning classes in each jar for the purposes of this analysis (Fig 3A). We detected an effect of temperature on cloning ($Z_{3,15}=4.464$; $p<0.001$). Post-hoc tests revealed that this effect was driven by the higher rate of cloning at 20°C ($Z_{3,15}=3.134$; $p<0.01$). Nevertheless, we did not see strong evidence for a positive trend in cloning across all temperatures ($R^2=0.1459$; $F_{1,13}=2.22$; $p=0.16$).

We then separated out the fully sized larvae in each treatment and replicate and haphazardly drew from only those for the morphometric and settlement analyses. We detected a jar effect in one of the 20°C replicates ($Z_{3,15}=-5.024$; $p<0.001$): this jar underwent particularly high cloning rates (apparently 100%), leaving no fully-sized larvae for the morphometric analyses. As such, the data presented in Fig 2 and S4 and S5 Figs for 20°C are based on the two replicate jars available for the analysis. For the settlement analysis (see below) we selected 10 larvae from the "regenerating clone" class (Class 2; see Methods) in this jar, since they were the most advanced larvae in the jar.

The morphometric results at the brachiolaria stage were somewhat different from the bipinnaria stage results. Here again we increased the number of features measured due to a series of new juvenile skeletal and other features present at the brachiolaria stage (Fig 1; S1 Table). We separated the characters into larval or incipient juvenile (including characters used only for the settlement transition, such as the brachiolar attachment complex). The larval characters had a similar optimal temperature as seen in earlier development, estimated to be 16.9°C (95% C.I.: 16.1–18.8°C; Fig 2, S4 Fig). However, we saw a very different pattern for the juvenile characters, for which we estimated the optimal temperature to

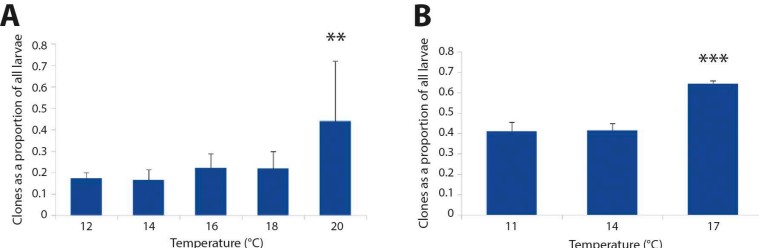

**Fig 3. Cloning rates increase at higher temperatures in brachiolaria stage larvae. (A)** *Exp 1*; **(B)** *Exp 3*. Asterisks indicate significant differences from other treatments (**p<0.01; ***p<0.001). Error bars are standard errors of the mean (s.e.m.).

be 14.0°C (12.0–14.8°C; Fig 2, S5 Fig). In other words, the larvae in warmer water appeared to be trading off preparation for settlement with continued somatic growth, whereas lower temperature larvae accelerated their ontogenetic trajectory towards settlement at the expense of larval growth.

This pattern was reinforced when we examined settlement patterns in larvae from this same brachiolaria-stage temperature exposure. Consistent with the juvenile growth data above, we saw a peak settlement response to a strong inducer [28] – fronds of the coralline alga, *Calliarthron tuberculosum* – at 14°C (Fig 4).

**Exp 2–Temp Shift Settle 2021**

**Distinguishing temperature effect per se from the impact of a temperature shift on settlement.** We note that the aforementioned 8–13 day larval temperature exposures in *Exp 1* (see Table 2) involved shifts from the culturing temperature (12°C) to the experimental exposure temperatures (12–20°C) at mid- or late larval stages. For the brachiolaria (late) stage exposures, the larvae grew at 12°C for 51 days until being shifted into their experimental treatments and replicates. As such we consider three hypotheses for the unexpected result that we described above for the lower temperature optimum at settlement for the brachiolaria stage exposures (Fig 4).

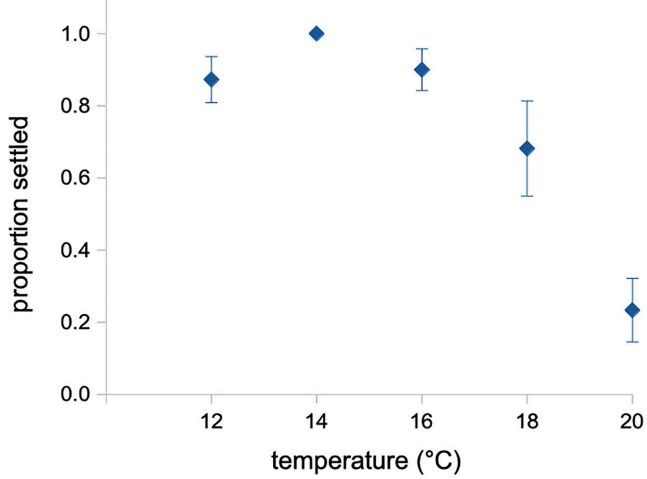

**Fig 4. Settlement in *Exp 1*.** Note that the relationship mirrors the results for juvenile characters in the brachiolaria stage, with the temperature optimum at 14°C. Error bars are s.e.m.

- **H1**: Larvae at a comparable developmental stage are more likely to settle at a lower temperature.

- **H2:** Larvae shifted from a lower to a higher temperature show inhibited settlement responses.

- **H3:** Larvae shifted from lower to higher temperature shift their developmental trajectories over the course of days towards larval and away from juvenile growth.

To address these hypotheses, we raised larvae at either 12 or 16°C throughout larval development, and then when they appeared competent, we shifted a subset to the opposite temperature. Immediately afterwards we examined their settlement patterns in response to *C. tuberculosum*, with 5 larvae per replicate exposure, and 6 replicates per temperature treatment (see Table 1 for summary).

H1 would predict that the 12°C reared larvae would settle more readily than the 16°C larvae at their respective temperatures. H2 would predict that the larvae shifted from 12 to 16°C would show immediate, inhibited settlement responses. H3 would predict no obvious differences among treatments in this short term temperature shift study.

The data in Fig 5 is most consistent with H3. Despite the subtle apparent effect, we observed no consistent differences in proportion settled due to rearing temperature (H1: $F_{1,20} = 0.072$; $p = 0.79$) or settlement temperature ($F_{1,20} = 1.040$; $p = 0.32$), and no interaction (H2: $F_{1,20} = 2.428$; $p = 0.14$). Overall, our results provide the most support for H3: that the settlement differences we observed in Fig 4 were due to a shift in investment towards larval growth – and away from juvenile growth – when larvae experienced an increase in temperature late in development (see Fig 2; Table 3).

### Exp 3–Settlement 2023

**Larvae grown at 17°C settle at a larger juvenile size than at 14°C or 11°C, and 11°C-reared larvae settle with fewer juvenile arms on average.** As a further exploration of the unexpected results we reported above for a shift in temperature optimum at settlement, we cultured larvae in 2023 in replicate jars (3 jars per treatment) at 11°C, 14°C and 17°C throughout larval development, and also added a treatment where we replaced half of the standard phytoplankton diet with natural plankton ('14-plankt'). In this experiment (see Table 1 for summary), we attempted to prevent spontaneous settlement by transferring the larvae to new, completely cleaned and dried jars at each water change.

At 63 dpf, we counted all remaining larvae in each of the 9 replicate jars: 3 each for 11°C, 14°C and 17°C. Note that we excluded the 14-plankt jars at this point, as these larvae were obviously less well-developed on average compared to any of the other treatments, a result that we ascribe to the lower than expected levels of plankton during most collection days during the larval rearing period (see Methods).

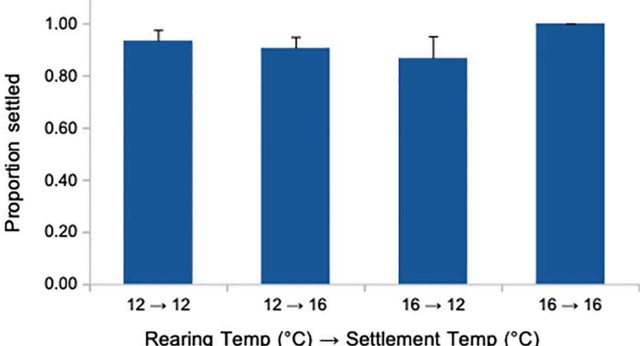

**Fig 5. No effect of rearing temperature nor of a pre-settlement shift in temperature on proportion settled in *Exp 2*.** Error bars are s.e.m.

We present the larval counts on 63 dpf in S4 Table, distinguishing fully grown competent larvae from various classes of smaller, mis-shaped and otherwise delayed larvae that we ascribe to cloning (see above). Note that the total numbers of larvae were different in the different jars on 63 dpf for three reasons: (1) different numbers of larvae had settled spontaneously in each jar before day 63; (2) the observed cloning not only changes the total numbers of larvae, but can result in loss of small clones through the mesh used for water exchanges; and (3) other sources of larval loss, including mortality and experimental loss during water exchanges. With those caveats, we detected an effect of rearing temperature on the proportion of larvae that were clones (Fig 3B). Specifically, rates of cloning were 50% higher at 17°C when compared to either 11°C ($Z_{3,9} = 7.518$; $p < 0.001$) or 14°C ($Z_{3,9} = 7.234$; $p < 0.001$).

After separating out all full-grown, seemingly-competent larvae (non-clones) in each replicate jar, we haphazardly selected 25 of these and exposed them to our standard strong settlement inducer (*C. tuberculosum*) in 200 ml jars. Note that for the 14-plankt larvae, which as mentioned were delayed relative to other treatments, we selected all seemingly-competent larvae for this settlement test, which amounted to fewer than 25 total 14-plankt larvae per replicate jar.

Then, after giving the larvae 11 days to fully complete their transformation to the juvenile stage, we measured the diameters of up to 12 haphazardly chosen juveniles per replicate jar (see Fig 6B for measurement method). We detected a strong effect of rearing temperature on juvenile diameter at settlement ($F_{3,12} = 25.26$, $p < 0.001$; Fig 6A, blue bars). Post hoc tests confirmed significant differences among all temperature treatments (11°C < 14°C < 17°C) and between 14°C-plankt and 17°C (see Fig 6A).

11 days after settlement, we also counted number of visible arms in juveniles and we detected a temperature effect ($F_{3,12} = 8.807$, $p < 0.001$; Fig 6A, orange bars), with 11°C-reared larvae settling with fewer arms than any of the other treatments (see Fig 6A). We have consistently noted a substantial proportion (approximately 10%) of juvenile sunflower stars settling with fewer than 5 arms (compare Fig 6 panels B,C); these juveniles can sometimes grow for weeks without forming additional arms, though in all cases, these juveniles eventually add arms. In one case, we had a three arm juvenile

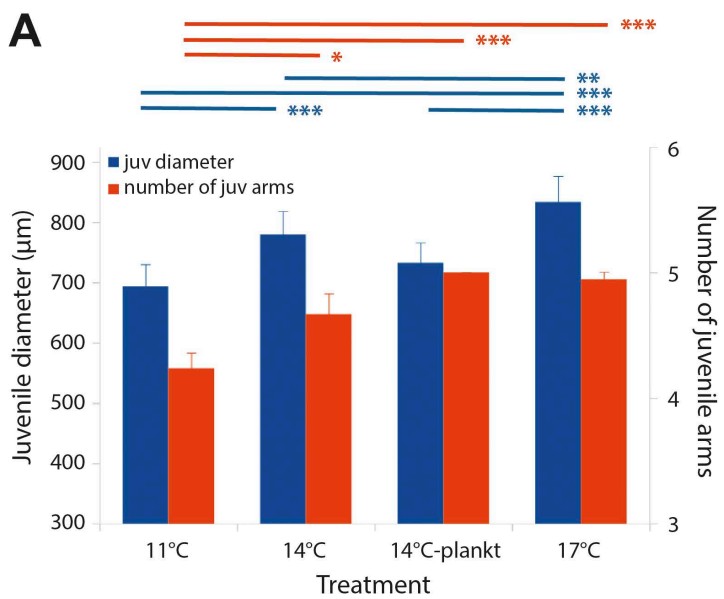
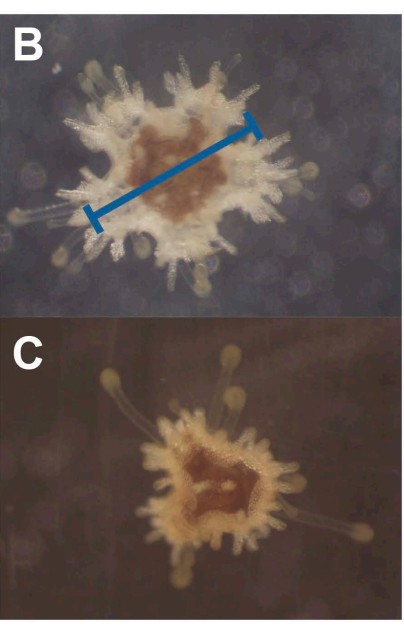

**Fig 6. Larvae raised at higher temperatures settle at a larger juvenile diameter and with more juvenile arms *(Exp 3)*.** (A) Juvenile diameter on left axis (*blue bars*); number of juvenile arms on right axis (*orange bars*). Lines (*above and color coded like the bars*) connect significantly different treatments in post-hoc tests: *$p < 0.05$; **$p < 0.01$, ***$p < 0.001$. Error bars are s.e.m. (B) 17°C juvenile with 5 arms, 780 μm diameter (*measurement method overlaid in blue*). (C) 11°C juvenile at the same scale with 4 arms, 590 μm diameter.

grow robustly for 70 days after settlement to 6 mm diameter (a 10-fold increase in diameter) before finally adding three additional arms simultaneously. We are unsure what to make of these aberrant arm numbers in general, and their connection to larval rearing temperature specifically, but offer some thoughts in the Discussion.

**Exp 4–Juv Growth 2022 and Exp 5–Juv Growth 2023**

**Newly-settled juveniles grow faster at 16–17°C than 11°C, with no increase in mortality, and no evidence for a larval-to-juvenile carry-over effect.** Here we set out to determine if the high temperature optimum in larvae extends to the juvenile stage, and if the larval rearing temperature influences the temperature at which juveniles grow best.

In *Exp 4–Juv Growth 2022*, we raised replicate batches of larvae at either 11°C or 16°C, and then tested the juveniles at either 11°C or 16°C in a 2x2 factorial design, with 12 juveniles in each of the 4 treatments (see Table 1 for summary).

30 of the 48 juveniles survived until the end of the 41-day experiment. Of the 18 that did not, 10 disappeared with no remains ('MIA'), 4 died after being stranded above the water line ('high and dry'), and 4 suffered bona fide mortality, 1 juvenile in each treatment ($F_{1,36} = 0.000$; $p = 1$), with no jar effect (larval rearing replicate: $F_{4,36} = 0.104$; $p = 0.31$).

'High and dry' mortality occurs because young juveniles apparently do not know to crawl down into the water when stranded above the water line as water levels fluctuate (Hodin, unpublished). Mortality in the 4 'high and dry' juveniles in the 2022 experiment showed no relationship to treatment ($F_{1,40} = 1.262$; $p = 0.27$). Unexpectedly, the high and dry condition showed a 'jar' effect: 3 of the 4 high and dry juveniles came from a single 11°C larval rearing replicate ($F_{4,36} = 3.0$; $p = 0.031$). And while there was no effect of the 12-chambered box in which the juveniles were reared ($F_{1,40} = 0.226$; $p = 0.64$), there was an effect of side of the box in which the juveniles were housed (hinge versus non-hinge side: $F_{1,40} = 4.520$; $p = 0.04$) but not chamber position (corner versus interior chamber: $F_{1,40} = 2.260$; $p = 0.14$). The way we arrange the flow into the chambers may have subtly impacted the manner (pitch) in which the boxes sat in the rearing incubators, thus leading to a greater likelihood of high and dry stranding on the hinge side of the rearing boxes.

In contrast with 'high and dry', there was a treatment effect to the 'MIA' juveniles – all 10 were in the high temperature treatments ($F_{1,40} = 15.067$; $p < 0.001$), 5 in each larval temperature batch (11--> 16 and 16--> 16; $F_{1,40} = 0.491$; $p = 0.62$). We are confident that these missing juveniles were due to clogging of the Nitex outflows with fouling diatoms specifically in the high temperature juvenile treatments (see Methods), resulting in chamber overflow and juvenile escape.

We assessed growth rates (µm day$^{-1}$) for each of the juveniles in 2022 as follows:

$$(D_T - D_0) / T \tag{1}$$

where T was the day of the experiment on which the final diameter measurement ($D_T$) was taken, and $D_0$ was the initial measurement before the experiment began (day 0). For the 30 juveniles that survived the experiment, T = 41; for the other 14 juveniles, T = the experimental day of the last measurement taken on that juvenile before it disappeared (MIA) or died (high and dry). We excluded the growth data for the 4 juveniles that suffered bona fide mortality (1 juvenile per treatment), all of which exhibited negative growth in the period before dying.

The growth data across treatments are summarized in Fig 7A. Juveniles reared at 16°C exhibited significantly higher growth than those reared at 11°C ($F_{1,41} = 4.579$; $p = 0.04$), with no significant effect of larval rearing temperature ($F_{1,41} = 2.506$; $p = 0.12$) and no interaction ($F_{1,41} = 0.091$; $p = 0.76$).

In *Exp 5–Juv Growth 2023*, we raised replicate batches of larvae at either 11°C or 17°C, and then tested the juveniles at either 11°C or 17°C in a 2x2 factorial design, with 12 juveniles in each of the 4 treatments (see Table 1 for summary).

In *Exp 5*, we were only able to run the full experiment for 2 weeks, because in the third week the 11°C temperature tank experienced a heat spike to 21°C for an unknown duration. All of the stars survived this heat exposure after returning to 11°C, but they were no longer suitable for experimental comparison. 47 of the 48 juveniles survived for the 2 weeks of the study.

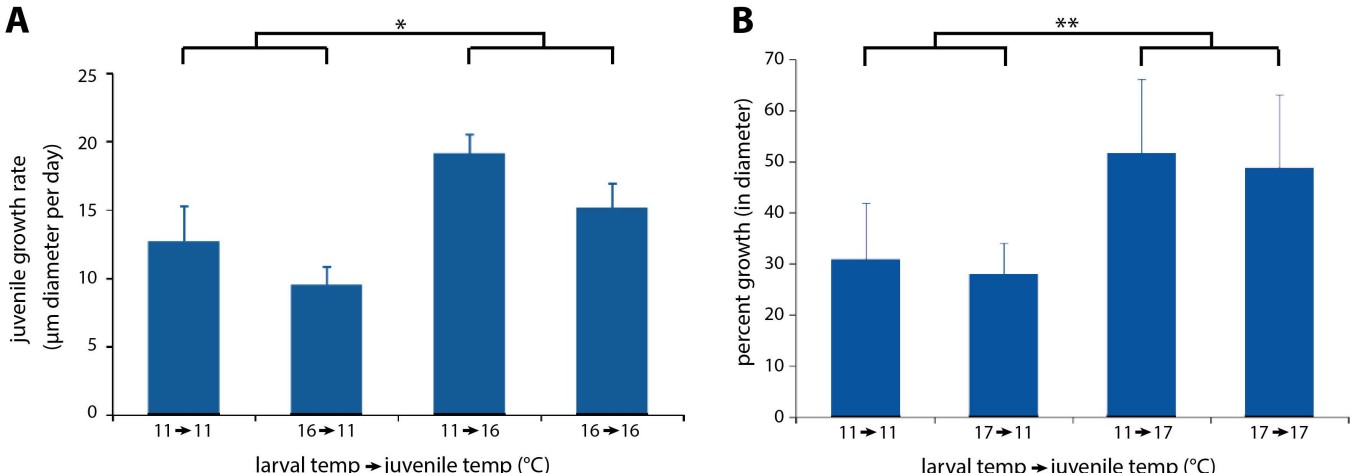

**Fig 7. Post-settlement juveniles grow faster at higher temperatures, irrespective of larval rearing temperature. (A)** *Exp 4* showing ~70% more rapid juvenile growth at 16°C whether reared as larvae at 11 or 16°C (*$p < 0.05$). **(B)** *Exp 5* showing ~70% more rapid juvenile growth at 17°C whether reared as larvae at 11 or 17°C (**$p < 0.01$). Error bars are s.e.m.

In *Exp 5*, we assessed growth rates for each of the surviving 47 juveniles in a similar manner as described for the 2022 data, above. But this time we simply expressed growth as percent growth in diameter over the two weeks of the experiment, namely:

$$(D_T - D_0)/ D_0 \qquad (2)$$

In this case, T was day 14 of the experiment for all juveniles.

The growth data across treatments are summarized in Fig 7B. Juveniles reared at 17°C exhibited significantly higher growth than those reared at 11°C ($F_{1,35} = 11.235$; $p = 0.002$), with no significant effect of larval rearing temperature ($F_{1,35} = 0.410$; $p = 0.5$) and no interaction ($F_{1,35} = 0.000$; $p = 0.99$).

Following the unanticipated 21°C heat spike in the 11°C juvenile treatments, the temperature comparison with 17°C was no longer valid, as stated above. Nevertheless, we continued following these juveniles, and can report that the 21°C heat spike juveniles all survived for at least a month (until we stopped following them individually) and grew at a comparable rate to the stars in the 17°C treatments ($F_{1,35} = 0.186$; $p = 0.67$). These anecdotal results again speak to the temperature resilience of sunflower star juveniles.

### Exp 6–Flipping 2023

**Juveniles 7–8 months after settlement show improved performance in a marine heatwave (MHW) simulation.** Righting behavior – timing how long it takes for a star on its aboral side to turn over, oral side down – is a standard protocol for examining echinoderm adult performance, including in response to environmental stressors [32–35]. It has been less often applied to echinoderm juveniles [36].

Here we report on a righting behavior ('flipping') protocol that we developed for assessing performance under heat stress. For our flipping protocol we video recorded individual stars for 10 minutes of continual flipping at a given temperature. Each star in the experiment went through the 10 minute regimen of flipping at ambient temperature (12°C), and then again, one week later, at the target temperature (12.0°C, 14.5 or 17.0°C). We compared the righting times for each star at ambient and target. The MHW and flipping protocol is diagrammed in Fig 8.

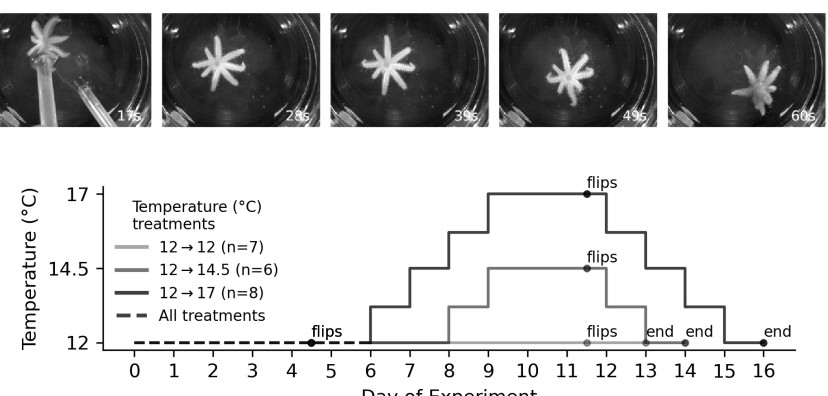

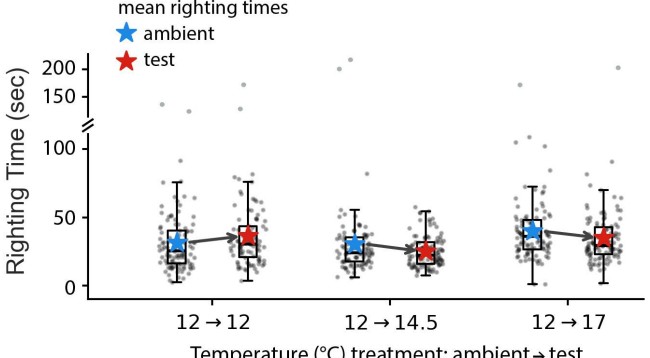

**Fig 8. Flipping protocol to assess righting time in juvenile sunflower stars (*Exp 6*).** *Montage at top:* Photo time lapse series showing a juvenile righting itself after being flipped by scoopula utensils in frame 1. This star was approximately 1.3 cm in diameter. *Graph at bottom:* Schematic of marine heatwave (MHW) simulation, with flip trials on day 4 (ambient, 12°C) and day 11 (target temperature).

The results are shown in Fig 9. The mean righting time in our ambient control trial was 34.5 ± 8.2 (95% c.i.) seconds. We detected no differences in righting time between the ambient and target temperature exposures in the 12°C control treatment ($Z_{6,195}$ = 1.486; $p$ = 0.137), but we detected a 18.8% decrease in righting time at 14.5°C ($Z_{5,192}$ = −2.428; $p$ = 0.015) and 10.3% decrease at 17°C ($Z_{7,208}$ = −2.089; $p$ = 0.037), suggesting improved performance at both of those temperatures relative to the control treatment. We detected no difference in righting time between 14.5°C and 17°C ($Z_{13,400}$ = −0.379; $p$ = 0.7).

We measured the diameters of each star at each flipping trial (ambient and test in the three treatments). Remarkably, the stars – which ranged between 0.9 and 1.8 cm in diameter at the start of the experiment – grew measurably during the one week period between the ambient and experimental tests (S6 Fig). The 12->12°C stars grew approximately 5% in diameter (growth of 0.7 ± 0.5 mm s.d.; $p$ = 0.023), the 12->14.5°C stars grew approximately 10% in diameter (1.1 ± 0.6 mm; $p$ = 0.047), and the 12->17°C stars grew approximately 11% in diameter (1.5 ± 0.6 mm; $p$ = 0.012). We did not detect any growth differences between the treatments by standard criteria (12->12 vs 12->14.5: t = −2.146 $p$ = 0.13; 12->12 vs 12->17: t = −2.632, p = 0.066; 12->14.5 vs 12->17: t = −0.204, p = 0.84).

**Fig 9. Juveniles showed improved righting time during a +2.5°C and +5.0°C simulated MHW.** Each point represents one flip for one star in each of the six flipping trials (*Exp 6*): three at ambient (*blue stars*) and three at the experimental test temperatures (*red stars*). Shown are standard box plots for each of the six trials with the arrows showing the trends in mean righting time for each of the three experimental treatments (ambient->test temperature): 12->12°C (n = 6 stars); 12->14.5°C (n = 7); and 12->17°C (n = 8). The positive trend at 12->12°C was not significant ($p$ = 0.13), whereas the negative trends at 14.5°C and 17°C were both significant ($p$ < 0.05; see the text).

## Discussion

The sunflower star (*Pycnopodia helianthoides*) is a generalist predator with a formerly broad geographic range from southeast Alaska to Baja California [10]. In 2013−14, its populations were devastated by an unprecedented outbreak of a still-mysterious disease known as seastar wasting (SSW [11]). In northern California, sunflower star disappearances have been linked to massive loss in kelp forests, due in large part to overgrazing by purple sea urchins in the absence of their sunflower star predators [22]. In the Salish Sea of Washington and further north, there are still remnant sunflower star populations, and it is from Salish Sea survivors that we initiated a conservation breeding program, and from which the larvae and juveniles used in this study were derived.

Due to the apparent critical importance of sunflower stars in maintaining kelp forest health [25], the hope is that this top predator can recover in California and elsewhere, and help reset the balance away from urchin barrens and towards kelp forests [26], either naturally or with human assistance. If sunflower star recovery is to occur, then offspring deriving from the remaining stars in the north of the former range, such as in the Salish Sea, would need to be able to thrive in the warmer waters in the south of their former range. In part to address whether this is possible, we set out to investigate the temperature resilience of key but poorly-studied early life stages of the sunflower star: embryos, larvae, and pre-reproductive juveniles.

### Early sunflower star life stages are remarkably robust to elevated temperatures

The common finding in all of our experiments – across multiple life stages and with diverse experimental techniques – is that Salish Sea sunflower stars show remarkable resilience to warm temperatures, far outside the range that they typically experience in the field.

In our embryo and larval studies (*Exp 1*), the sunflower star optimal growth temperature was 16–18°C, which is 2–4°C higher than the typical maximum (summertime) near surface temperature in our region (~14°C) [37–39]. Based on typical environmental conditions that the larvae would experience, this high temperature optimum was especially unexpected. Our spontaneous spawning observations and seasonal gonad analyses [28] indicate that sunflower star larvae are in the plankton predominantly in the winter and spring when Salish Sea temperatures are 10–12°C, which is 4–8°C below the optimal larval temperature according to our studies. Even summertime marine heatwave (MHW) events in our region rarely result in subtidal temperatures above 16°C [39,40].

When we raised larvae at 11, 14 or 17°C (*Exp 3*), we observed larger initial juvenile diameter at the highest temperature tested, and evidence for underdeveloped juveniles at the lowest temperature (fewer arms at settlement on average). In this sense, these results on initial (post-settlement) juvenile morphology mirrored the embryo and larval results recounted above.

Our studies of growth and survival of newly-settled juveniles compared two temperatures in two different cohorts: ambient (11°C) versus elevated (16–17°C). In both experiments (*Exps 4–5*), juveniles grew faster and survived equally well at the elevated temperatures, again speaking to sunflower star temperature resilience in comparison to typical Salish Sea temperatures. We did not detect any carry-over effect (sensu [41]) of larval rearing temperature on post-settlement growth or survival.

Finally, we tested the performance of 7–8 month post-settlement juveniles to a simulated MHW (*Exp 6*), using righting time as our metric of performance: the oral side up position is a vulnerable one, and hence the animals are motivated to restore the aboral side up position as quickly as possible [32]. While righting time is a widely-used index for performance in different echinoderms (e.g., [33–35]), it has been rarely employed in juveniles (but see [36]). We observed a mean 14.6% increase in performance (faster righting) at 14.5°C and 17°C when compared to the ambient test temperature (12°C). We note that sunflower star juveniles are very adept at righting themselves, with their mean righting time being about 6 times faster than that reported for juveniles of another multi-armed star: the Crown-of-Thorns sea star, *Acanthaster planci* [36].

Taken together, these results suggest that over the first 9 months of ontogeny, spanning the key planktonic-to-benthic settlement transition during metamorphosis, sunflower stars perform best at warm temperatures. In fact, their optimum temperature is several °C warmer than typical peak summertime temperatures in the Salish Sea, and approximating shallow subtidal temperatures during extreme MHW conditions [39], such as the heat dome of Summer 2021 (see [42]).

Our conclusion is that for these critical early life stages, sunflower stars demonstrate not only significant robustness to current and predicted near-future temperatures in the Salish Sea, but that Salish Sea sunflower stars could possibly be a viable source for repopulating waters in the south of their former range – namely Oregon and California, USA and Baja California, México – either by natural larval transport or ex situ breeding and rewilding.

We note emerging evidence of overall lack of population genetic structure across the former sunflower star range, from Alaska to California (L.M. Schiebelhut, pers. comm.). These results are consistent with recent historical gene flow across this range, presumably via larval transport (see below). In that sense, our findings on temperature robustness of Salish Sea sunflower stars may reflect the historical population dynamics, in that Salish Sea genotypes appear well adapted to California temperature conditions as well.

We further note that our results should not be taken as contradicting prior reports of a connection between temperature and severity of SSW [43]. In our own sunflower star lab colony, we have noticed that stars exposed to wasting have a better chance of survival at lower temperatures, consistent with published data on the ochre star, *Pisaster ochraceous* [44]. Our results presented here suggest that the connection between temperature and wasting may have less to do with stress on the star per se, and instead might point to temperature impacts on the disease-causing agent itself or its dynamics in the wild.

One final caveat is that most of our experiments were necessarily conducted with other organisms present: unicellular algae fed to larvae, coralline algae for settlement, and live prey in juvenile assays. Thus, we cannot fully rule out interaction effects between sunflower stars and these organisms across temperatures. For instance, improved juvenile growth at higher temperatures may partly reflect weakened prey, making them easier to capture. While our experiments did not directly test this possibility, we argue such effects do not undermine our conclusions. Our trial arenas paralleled the natural context in which predators and prey coexist; in these arenas, our stars performed best at higher temperatures. Additionally, results from our experiments without added organisms (e.g., the embryo growth and righting assays) aligned with those that included them, supporting the conclusion that the high-temperature optimum is truly inherent to early life-stage sunflower stars.

## Cloning increases at warmer temperatures

Larval cloning is an apparently widespread phenomenon in echinoderms [30], especially sea stars. We had previously reported observing significant proportions of clones throughout larval development in *P. helianthoides* [28]. In our experiments reported herein, we once again observed larval cloning throughout their planktonic stage, and in two experiments – *Exp 1* and *Exp 3* (see Table 1) – we did thorough counts of all larvae in all treatment jars to examine any evidence for an effect of temperature on larval cloning.

In both experiments, we saw a strong effect of temperature on cloning, with the highest temperature tested (20°C in *Exp 1*; 17°C in *Exp 3*) showing increased rates of cloning. A caveat for the *Exp 1* result is that this increase at 20°C was driven almost entirely by a single replicate jar at 20°C that exhibited 100% cloning. If we hypothesize that cloning larvae signal other larvae to clone, then an anomalous increase in cloning could be amplified, leading to the temperature effect in that singular jar. By contrast, the effect in *Exp 3* was consistent at 17°C across replicates. This raises the possibility that – like the reported increase in cloning with increased food availability [31,45,46] – the increase in cloning at 17°C is a response to ideal planktonic conditions, beneficial for rapid growth, and hence an advantageous milieu for cloning.

In contrast with our findings, Vickery & McClintock [45] reported that cloning occurred in late stage ochre star larvae at medium (12–15°C) but not low (7–10°C) or high (17–20°C) temperatures. In these ochre star experiments, 12–15°C was

determined to be the optimal growth temperature, again suggesting a correlation between optimal planktonic conditions and cloning.

## Temperature effects on post-settlement size and morphology

One remarkable feature of sunflower star larvae is that their planktonic duration can be extremely long. We have raised larvae for over 5 months and still settled them successfully into viable juveniles. Others have maintained viable cultures of sunflower star larvae for over 10 months (A. Kim, pers. comm.). In the absence of a settlement cue, competent sunflower star larvae can continue to feed and develop, during which time the rudiment grows more prominent [28]. Larvae with more prominent rudiments produce larger juveniles at settlement (JH unpublished observations), all of which suggests that delays in settlement could have a potential benefit for post-settlement juvenile growth. If true, this adds nuance to the idea that the intense predation pressure on planktotrophic larvae – in addition to other perils of extended dispersal [47,48] – impels them to depart the plankton as soon as possible [49,50]. While such predation can certainly be intense [51], the dangers for larvae may be traded off against greater post-settlement growth and survival in larvae that delay settlement [52].

In the studies reported herein and previously [28], we noted significant variation in size (juvenile diameter) and morphology (number of visible arms) at settlement both within and among larval cultures. Here we found that such variation is temperature dependent: larvae reared at temperatures below their growth optimum yielded smaller juveniles with fewer arms at settlement when compared to larvae reared at the 17°C, close to the larval growth optimum. We suspect that this example of phenotypic plasticity may be best understood if we consider it alongside a different and surprising instance of phenotypic plasticity that we observed specifically at late larval stages.

## An unexpected instance of plasticity to temperature in late stage larvae

We only report on one result that is not fully consistent with the basic conclusion of high temperature optimum in early life stages of Salish Sea sunflower stars. In our larval experiment on advanced larvae (*Exp 1*), we detected a temperature based trade-off between growth of larval structures (e.g., larval body length, stomach length) and growth of incipient juvenile structures (e.g., skeletal elements in the juvenile rudiment, settlement attachment complex). This trade-off manifested as distinct temperature optima for the two classes of growth: a larval growth optimum at 16–18°C and an optimum for incipient juvenile structure development at approximately 14°C. In other words at higher temperatures, the larval structures grew while juvenile structures were delayed, whereas at lower temperatures, growth of juvenile structures was accelerated relative to larval structures.

This result is reminiscent of a well-described trade-off in echinoderm and other invertebrate larvae in response to larval food levels: in high food, juvenile growth accelerates at the expense of larval structures, whereas at low food juvenile structures are delayed and larval structures grow preferentially [29]. The typical adaptive explanation for this result follows from the assumption that the plankton is a dangerous place, and under high food conditions, the larva is motivated to settle earlier and shifts its ontogenetic trajectory towards that end. By contrast, under low food conditions, there is insufficient food to fuel rapid growth through metamorphosis, so the larva grows a larger body, and hence a longer ciliary band for more efficiently capturing the less concentrated food particles under those conditions [53].

But why would temperature mimic (i.e., phenocopy) such an effect? We offer several possible explanations. First is simply that the high temperature conditions are mimicking low food conditions due to the higher metabolic demands of the larvae under high food. In our experiments we feed all larvae the same amount every two days, but during the intervening period, the higher temperature larvae fed more (due to aforementioned higher metabolic demands, as well as their larger bodies and hence greater particle capture ability), and thus cleared out the food in the culturing jar to a greater extent. Our anecdotal observations suggest that the guts of the higher temperature larvae are lighter than the low temperature larvae at water changes, suggesting episodic food limitation during their 48 hr feeding regimen.

We do not favor this explanation for the observed plasticity for two reasons. First, because one typically has to drastically reduce food levels (e.g., 10-fold or more as in [54]) to detect the degree of plasticity that we observed here. And second, because we saw the differences in gut color throughout larval ontogeny, but only observed the trade-off at the very end of larval ontogeny.

Another possible explanation for the tradeoff is that the observed plasticity is an adaptive response of high food larvae to optimal planktonic growth conditions, making the larvae less "desperate" to depart the plankton [49,52]. As noted previously, we have observed that delaying settlement can lead to larger juveniles at settlement, which can correlate with better initial benthic performance. We also reported herein that culturing larvae throughout the planktonic period at the temperature optimum of 17°C results in juveniles settling more fully developed and at a larger size than at cooler temperatures (*Exp 3*).

The key difference between *Exp 3* and the experiment showing the larval versus juvenile character trade-off (*Exp 1*) was that *Exp 1* involved a temperature shift late in ontogeny from 12°C to the target temperature (12, 14, 16, 18 or 20°C; see Table 2). This methodological difference between *Exp 3* and *Exp 1* indicates that the observed trade-off in *Exp 1* was a response to the temperature shift rather than to the temperature itself. As such, a temperature shift late in larval development may shift larvae into a growth trajectory that would result in delayed settlement, yet ultimately improved performance in the benthos when they arrive there.

A third explanation is a non-adaptive one. The protein machinery involved in building the larva simply has a higher temperature optimum than the protein machinery involved in building the juvenile. Our mid larval (bipinnaria; *Exp 1*) and early juvenile results (*Exps 4,5*) argue against this possibility, since in each of those stages, the optimal temperature for juvenile (or incipient juvenile) growth was 16–18°C. Nevertheless, we cannot exclude this explanation based on our experiments alone.

One consequence in *Exp 1* of the more rapid growth of juvenile structures in late stage larvae at 14°C was that a greater proportion of juveniles subsequently settled in those trials when compared to the 16–18°C juveniles. We caution the readers that we do not consider this finding to be supportive evidence for optimal settlement at 14°C. First, the higher temperature treatment larvae did eventually attain competence to settle, and settled normally. In that sense we did not observe more successful settlement at 14°C, we observed settlement at an earlier date. Furthermore, it should be noted that the settlement result was predicted, as we know that accelerated growth of juvenile structures in echinoderm larvae leads to earlier attainment of competence (e.g., [55]).

Furthermore, our experiment rearing larvae throughout all of larval development at 11, 14 and 17°C (Exp 3) revealed optimal settlement – as evidenced by size and morphology – at 17°C.

Further study is needed to deepen our understanding of the implications of this unexpected temperature based larval plasticity, and speaks to our need to investigate the broad ecological context in which these larvae find themselves as they are making their most consequential decision on whether or not to settle [56]. Furthermore, we need to recognize that such plasticity may not be adaptive; demonstrating that the plasticity is adaptive requires further study as well.

With those caveats in mind, we can confidently conclude the following about this unexpected late-stage larval plasticity in sunflower stars: it is indicative of a genome that can respond to temperature by modifying ontogeny in complex ways [57]. Such an ability – alongside the high temperature resilience that we report herein – seems to us to be a beneficial feature, particularly for an endangered species in the current epoch of accelerating anthropogenic climate change.

## Methods

### Broodstock, collecting, spawning, fertilizations

Adults (>30 cm arm tip-to-arm tip diameter) of *Pycnopodia helianthoides* were collected at various intertidal and subtidal sites in and around San Juan Island, WA (USA) in Spring and Summer 2019 (see [28]; Table 4). The director of Friday Harbor Laboratories (FHL) approved these collections under the auspices of state statute (House Bill 68,

R.C.W.28.77.230, 1969 Revision R.C.W.28B.20.320), with FHL as the managing agency. Collected adults were held in aquariums with constantly flowing, well-oxygenated natural sea water, and fed a diet of primarily mussels (~60–120 g wet) every two days.

In a previous publication [28], we reported limited success with sunflower stars using the standard method to induce sea star spawning [58]: injection of 1-methyladenine (1-MA). Since that time, we have consistently had success with this method. We believe that the discrepancy in the 2021 study was due to injecting stars outside of their peak reproductive season, which we found to be January-April. This contrasts prior reports that sunflower stars mainly spawn during spring-summer in our region [59–61].

We conducted the experiments reported herein across three years (2021–2023), with one set of fertilizations yielding cultures deriving from mixtures of individual male-female crosses each year (see [28]; Table 4). In 2021 (*Exps 1–2*), we obtained gametes by the previously-reported method of arm dissection, followed by maturation of oocytes *in vitro* using 1 µM 1-MA treatment [28]. In 2022 and 2023 (*Exps 3–6*), we obtained gametes by injection of 1 ml of 100 µM 1-MA per 100 ml of star volume (based on the volume of water the star displaces when completely submerged). We injected half of the total volume of 1-MA into the coelomic cavity of the star at the base of each of two arms on opposite sides of the star, avoiding regenerating arms. In gravid males this triggered spawning around the entire circumference of the star, whereas injected gravid females often spawned only adjacent to the injection sites, at least initially. Mean (± 95% c.i.) time from injection to spawning at ambient winter water temperatures (8–9°C) in 2022–2023 was 110 (±20) min in males and 183 (±9) min in females.

While we prefer the 1-MA injection method as being less invasive, either method is viable. Our observations indicate that one can obtain fertilizable gametes slightly in advance of the peak reproductive season (i.e., in Nov-Dec) by dissection, at a time of year when the females do not seem to readily spawn in response to 1-MA injection (Hodin et al 2021). We have not observed any consistent association of seastar wasting (SSW) with either arm amputation or 1-MA injection in our broodstock stars from 2019 to 2024. However, we do periodically detect outbreaks of SSW in our colony in the winter and spring, suggesting a possible association between the reproductive cycle and sensitivity to wasting.

From injected stars, we collected sperm as 'dry' as possible (i.e., with minimal seawater) using a Pasteur pipette, and stored it in Eppendorf tubes at 4°C until fertilizations. We collected eggs using a turkey baster or plastic transfer pipette from the direct vicinity of the gonopores, and held them in beakers of 1 µm-filtered natural sea water (MFSW) at ambient temperatures until fertilizations, within 6 hrs of collection. See [28] for gamete preparation protocols following arm

**Table 4. Details of fertilizations, parentage, and cross design for the embryo, larvae and juveniles studied herein.**

| Fertilization date(s) | Names of parents (M or F) | Collection month and locations (all off or near San Juan Island WA) | Design |
|---|---|---|---|
| 01/20/2021 | Isabella (M) | 07/2019 Snug Harbor intertidal | 2M x 2F (4 crosses mixed) |
| | Stella (M) | 06/2019 Friday Harbor (FH) subtidal | |
| | Gaucho (F) | 06/2019 Cattle Point intertidal | |
| | Prospero (F) | 08/2019 Brown Island subtidal (crab pot) | |
| 01/02/2022 | Clooney (M) | 04/2019 FH Labs (FHL) dock | 2M x 1F (2 crosses mixed) |
| | Stella (M) | 06/2019 FH subtidal | |
| | Van Gogh (F) | 06/2019 FH subtidal | |
| 01/26/2023−01/27/2023 | Olga (M) | 08/2019 Snug Harbor subtidal | 3M x 2F (6 crosses mixed) |
| | Harriett (M) | 07/2019 FHL dock | |
| | Crocus (M) | 07/2019 Beaverton Cove intertidal | |
| | Fulgens (F) | 05/2019 FHL dock | |
| | Deep Blue (F) | 07/2019 Snug Harbor subtidal | |

Note that all experiments in a given year came from fertilizations that same year.

amputation. In Table 4, we list the parentage and fertilization dates of the 2021, 2022 and 2023 fertilizations used to generate the embryos, larvae and juveniles for this study. Fertilizations followed the methods detailed in [28].

## Embryogenesis through early juvenile stages, general methods

We cultured embryos and larvae as described in [28], but using a single large motor-driven stirring apparatus [62] suspended over glass jars in a series of rectangular plexiglass tanks (Fig 10). Each tank had independent temperature control as described below. The result was that all larval cultures (jars) across temperature treatments were raised under the same environmental conditions and stirred in a consistent manner, thus avoiding pseudoreplication in rearing conditions across temperature treatments. Every 2 days, we changed ~95% of the culture water by reverse filtration (2021–22) or 200% of the culture water by forward filtration (2023), using micron mesh filters that were no more than 1/2 the width of a given larval stage (see [63] for details on these two filtration methods). We hand-cleaned cultures as needed and fed the larvae a standard diet of 2.5 cells/µl *Rhodomonas spp.* and 3.0 cells/µl *Dunaliella tertiolecta*. Initial larval densities were < 1 larva/ml. After approximately 2–3 weeks of larval culture, we thinned out the cultures gradually over two water changes to a final density of approximately 0.2 larva/ml. We chose this time frame because it is when the left and right coeloms of the larva fuse both anterior to the mouth and posterior to the stomach, an indication that the development of the juvenile rudiment is about to begin (see [63]).

When larvae reached apparent metamorphic competence (see [28]), we induced settlement in wide-mouth, half-pint glass canning jars (Bell) that were coated with a monolayer biofilm of two benthic diatom species: *Navicula incerta* and *Nitzschia frustulum.* These benthic diatoms make a suitable substratum to support juvenile growth and help prevent fouling by opportunistic diatom species that can be lethal to juveniles (see [28]). We pre-filled the half-pint jars with 200 ml MFSW at the desired temperature and added 2.5 g of temperature-acclimated fronds of the articulated coralline alga, *Calliarthron tuberculosum*, as a settlement inducer to each jar [28]. We then added up to 50 larvae to each jar,

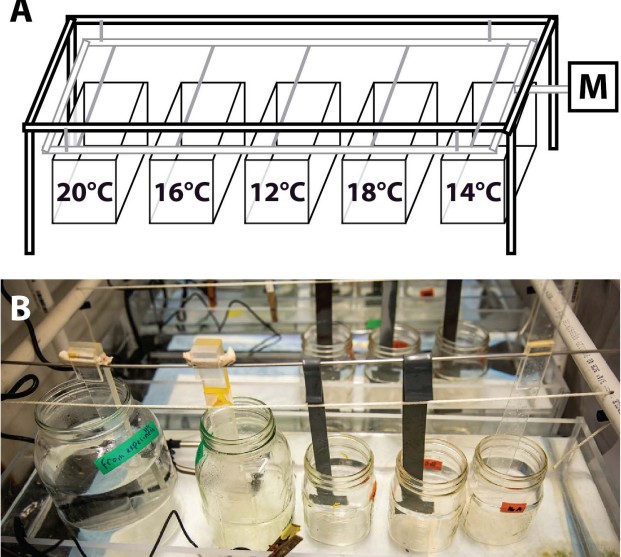

**Fig 10. Temperature-controlled larval stirring apparatus used throughout the study. (A)** A rotary motor (*M*) causes a hanging platform to oscillate, moving paddles that stir the jars (see panel B) that are sitting in the temperature controlled plexiglass tanks (*labeled with temperature*). **(B)** Photo of the apparatus with quart, half gallon and gallon culturing jars being stirred (Dennis Wise/ UW Media).

thus maintaining consistent larval densities from the larval cultures to the settlement jars. After 48 hrs, we removed any non-settled larvae, and performed daily 100% water changes for the settled juveniles in the jars until the beginning (see below) of our juvenile temperature exposures.

## Temperature manipulations

**Embryos and larvae.** We conducted all larval culturing in a single multi-temperature rearing system illustrated in Fig 10, in a 10°C cold room. We raised larvae in quart, half gallon and gallon wide-mouthed glass canning jars (Bell), with each jar as a replicate and multiple larvae per jar. Each of the five rectangular plexiglass chambers shown in Fig 10B was a 16 liter water bath, heated above ambient using a 100W titanium heater (Bulk Reef Supply) controlled by a programmable temperature controller (Inkbird model ITC-306T) set with a 0.1°C drift allowance. A submersible pump kept water circulating in each chamber to maintain temperature consistency throughout the chamber, and hence the constantly-stirred culture jars therein. We randomized the order of temperature treatments (i.e., the position of the 5 temperature chambers) relative to the front of the stirring rack in an attempt to avoid any bias in growth as it may relate to differences across the stirring rack (e.g., incident light). During manipulations such as water changes, larvae were maintained within ±~1°C of their rearing temperature at all times.

**Post-settlement juvenile growth (*Exps 4–5*).** We conducted all growth experiments with recently-settled juveniles, isolated in individual chambers within a 12-chamber box. With the use of a 12-line irrigation splitter fed from a submersible pump, each chamber (8 x 4 x 2 cm; 64 ml volume) had an inflow at the top and an outflow through 150-μm Nitex mesh at the outer edge, resulting in a ~100% water change in each chamber on average every 75 seconds. We fed the juveniles ad libitum with newly-settled <0.5 mm diameter purple (*Strongylocentrotus purpuratus*) or green (*S. droebachiensis*) sea urchins, which we reared using our standard urchin rearing methods (see [64]) except that we used a more efficient forward filtration method for water changes in 2022–2023 (see [63]). The 12-chamber boxes were pre-incubated for 2–3 days at room temperature (~20°C) under grow lights with a 50/50 mixture of two aforementioned beneficial benthic diatoms – *Navicula incerta* and *Nitzschia frustulum* – to provide a suitable substratum for the sunflower stars (see [28]) and food for the sea urchin juvenile prey. The boxes containing juveniles were then partly submerged in temperature controlled coolers (~100 L) in the FHL Environmental Research Lab, maintained at approximately ± 0.1°C variance from the target temperature (see Table 1) at all times.

In this system in 2022 (*Exp 4*), we used sand filtered sea water on slow flow-through (~1 L min⁻¹), resulting in a full water exchange in the coolers every ~2 hours. An unintended result of filtration with only the sand filter was that the coolers (especially the higher temperature treatments) became fouled with a problematic, opportunistic assemblage of benthic diatoms that we have observed to be harmful to juveniles [28] and also clogged the Nitex mesh, hindering flow out of the chambers. Therefore, we changed coolers every two weeks in an attempt to limit such fouling. In 2023 (*Exp 5*), we installed micron filtration (stepped down to 1 μm nominal) upstream of the coolers, under the same flow rate. These latter steps limited harmful diatom assemblage accumulation as well as the Nitex mesh clogging.

**7-8 month post-settlement juvenile performance (*Exp 6*).** For *Exp 6*, we cultured individual juveniles in experimental rearing chambers (as described further below) for 7–8 months until the initiation of experiments, feeding them an ad libitum diet of juvenile sea urchins (*S. purpuratus* and *Mesocentrotus franciscanus*), juvenile oysters (*Magallana gigas*) and juvenile Manila clams (*Ruditapes philippinarum*), where the prey's test or shell diameter was always < 0.75x the predator diameter.

We cleaned and fed boxes and cages weekly, and transferred stars to new cages or boxes every other week. The ambient conditions varied due to our main juvenile rearing tank being on flow-through natural MFSW during this 6 month period. Sea table temperatures varied from 10°C in the spring, to ~14°C during the summer, and down to 8°C when the experiments were undertaken in Nov and Dec 2023. Ambient salinities, pH and oxygen saturation also varied from mean (and 95% most common range) of approximately 31.2 psu (28.7–31.4 psu), 7.79 (7.58–8.06), and 73% (55–146%)

respectively [39]. During summer 2023, daytime heat spikes in our aquaria occasionally reached 16°C (about 2°C above ambient) due to the FHL outdoor sea water lines warming in the sun.

Temperature acclimation and ramp-up in this experiment took place in a shallow, slowly flowing sea table with nominal 20 μm FSW in which we placed two 100 W titanium heaters and recirculating pumps, with temperature controlled by the same Inkbird system mentioned in the prior section. This system maintained SW temperatures with 0.1°C of the target.

## Experiment-specific methods

***Exp 1–Larv 2021.*** The goals of this experiment were to assess the growth and survival of the planktonic stages of sunflower stars across a range of temperatures, and to determine if their temperature optimum changes as larval ontogeny proceeds. We conducted short term (5–13 day) temperature exposures across an 8°C range of temperatures (in 2°C increments) at three different planktonic stages: post-hatching embryogenesis (5 days), early larval development (bipinnaria stage; 8 days) and late larval development (brachiolaria stage; 13 days).

The parentage and fertilization details for this experiment are shown in Table 4. We raised post fertilization cultures at ambient temperatures (~9°C) until hatching (48 hrs post fertilization; hpf), at which time we combined embryos from all four single M x F crosses in equal proportions, and then haphazardly drew from that mixed culture to set up the embryogenesis study. The larvae for the experiment were from 'mother cultures' from the same 2M x 2F set of crosses, cultured at ambient (12°C) until the onset of the bipinnaria and brachiolaria experiments respectively.

At the start of each stage of study, we transferred embryos or larvae (see Table 2 for numbers) into each of 15 replicate wide-mouth glass jars containing 600 ml MFSW at 12°C, and then randomly assigned each jar to one of five temperature treatments, 3 replicate jars per treatment. Then we placed the jars into their respective temperatures in our multi-temperature rearing system (Fig 10) and stirred them with individual paddles [62], allowing them to warm or cool gradually to their target temperatures. From that point until the end of each 5–13 day exposure those embryos or larvae were maintained at their experimental temperature. The details for each of the three studies (exposure temperatures, dpf started and ended, and dates of data sampling) are shown in schematic in Table 2.

At periodic intervals (see Table 2), we haphazardly picked 15 larvae from each jar to mount on a microscope slide with raised cover glass [62]. Using an AmScope MC300 camera mounted on a compound scope (Olympus BH-2), we then photographed the first 10 larvae encountered on each slide, only skipping those that were at an unfavorable orientation for photographing (e.g., larva obscured by being under edge of cover slip), were presumed results of cloning (see below), or that were distorted by compression. We used photographs of a stage micrometer taken at the same magnifications for scale. As described further below, we used these photos for embryo and larval morphometrics. In the two larval experiments, we also characterized the growth of juvenile features in each live-mounted larva as detailed in [28], Fig 1 and S1 Table. This latter scoring in the bipinnaria stage exposure was conducted blind (i.e., the person doing the scoring was unaware of the temperature treatment corresponding to each microscope slide); this was not possible in the brachiolaria stage exposure, due to the unavailability of an assistant on the day of scoring.

We conducted embryo and larval morphometrics from the photos using Fiji (ImageJ). In order to avoid unintentional bias, the person doing the measurements was unaware of the treatment and replicate corresponding to each photo (i.e., the larvae were measured 'blind'). Fig 1 shows the features we measured at each stage, with further details in S1 Table. We would bypass an embryo or larva if it was determined by the person measuring to have been compressed laterally under the cover glass, thus distorting the overall shape. In some cases, this resulted in fewer than 10 larvae to analyze. Furthermore, if an embryo or larva's orientation obscured a given feature, the measurement for that feature was left blank by the blind scorer.

In both larval stages, but especially in the longer brachiolaria stage exposures, we recorded substantial proportions of cloned larvae, as expected [28,31]. Because clones are often missing large portions of the larva, morphometric

measurements would not be appropriate. Before the brachiolaria analysis, we examined each larva in a dissecting scope and characterized it as Class 1–4 as follows:

- **Class 1:** complete larva, comparable size of the larvae to the initial size at selection, 13 days earlier.

- **Class 2:** mostly complete larva, but <75% initial size indicating that cloning had occurred, followed by some regeneration.

- **Class 3:** small clones, <50% initial size, thus clearly either the smaller half of a cloning event or something that cloned in half or more than once.

- **Class 4:** one of a variety of other tiny clones, <20% initial size.

We counted and analyzed the distribution of these classes (as well as the small number of spontaneously settled juveniles) across treatments and replicates, but only used Class 1 larvae from each jar for morphometrics and settlement tests.

We conducted the settlement tests in 6-well multiwell plates that had been incubated in tanks with adult *P. helianthoides* for 7 days to accumulate an inductive biofilm [28]. Then, after gently rising out the wells, we filled the wells with 8 ml MFSW and added approximately 0.1 g (~10 segments) of freshly collected fronds of the coralline alga *Calliarthron tuberculosum* – a potent settlement inducer [28] – to each well, and placed the plate into the larval rearing system to equilibrate the algae and biofilm to the target temperature for 24 hrs, one plate per temperature treatment.

At 64 days post fertilization (dpf) in the brachiolaria stage exposures, we haphazardly selected 30 Class 1 larvae from each replicate jar into a 125 ml beaker, and with the naked eye to avoid unconscious selection, haphazardly picked 10 of the 30 larvae for settlement tests, placing 5 into each of 2 wells of the appropriate temperature-equilibrated well plate. Then, from the remaining 20 larvae in the beakers we haphazardly picked 15 larvae to be mounted on a slide for morphometrics and staging, as detailed above.

We exposed larvae in the settlement tests at their experimental temperature (12, 14, 16, 18 or 20°C respectively) for 19 hours and then individually assessed them (per [28]) as either unattached larvae (swimming or touching a surface of the well or alga but not attached), attached (with their brachiolar arms but not settled), or settling (in the process of irreversible settlement and contraction of larval tissues). Larvae were then maintained at their experimental temperatures, allowed to complete settlement for 5 more days, and then assessed one final time as either settled or not.

**Exp 2–Temp Shift Settle 2021.** To control for the possible effects of shifting temperature (rather than the exposure temperature per se) on settlement, we undertook a second experiment in 2021 where we raised larvae at either 12°C or 16°C from 9 dpf until 84 dpf, and then compared their settlement responses at either 12°C or 16°C, in a fully factorial design. Crosses and fertilization dates are listed in Table 4; other experimental details are summarized in Table 1. Larvae in these jars started settling spontaneously on the sides of jars at about 50 dpf, but we were able to stave off settlement in the majority of the larvae through thorough jar cleaning, and hence biofilm removal, every other day.

On 84 dpf, most of the remaining larvae in the jars appeared competent, so we haphazardly selected 30 competent-looking larvae (well developed 'helmets' and brachiolar apparatuses; see [28] and S1 Table) from the 12°C cultures into each of two 125 ml beakers of 12°C MFSW, and did the same for the 16°C treatments. Then for each pair, we flipped a coin to assign one beaker from each rearing temperature to a 12°C settlement treatment and one to a 16°C treatment, at which point we moved each beaker to its target temperature to allow 30–45 minutes for acclimation.

Then we transferred the 30 larvae haphazardly into six wells of a 6-well multiwell plate, 5 larvae per well. These plates had been prepared as described for the prior experiment, with biofilm and then a 24 hr temperature acclimation for the *C. tuberculosum* fronds. We scored them 24 hrs later as described in the prior experiment. We maintained these juveniles for 3 weeks and scored them one final time to assess post-settlement survival.

**Exp 4–Juv Growth 2022.** Crosses and fertilization dates for this experiment are listed in Table 4; other experimental details are summarized in Table 1. We used the same methods as in 2021 except as follows. We raised 3 replicate jars of 200 larvae each through larval development at either 11 or 16°C. For settlement, we prepared six 200 ml jars with biofilm for one week as described above for the well plates, added 2.5 g *C. tuberculosum* per jar, and placed three jars at 12°C and three at 16°C for a 24 hr equilibration period. Then on 59 dpf, we selected up to 50 competent looking larvae from each of the six replicate jars for settlement (one 11°C replicate and one 16°C replicate had fewer than 50 competent larvae, so we selected 43 and 26 larvae respectively from these jars), and transferred them to a settlement jar, one jar per replicate, at the larval rearing temperature.

After 7 days, we selected 8 normally-developing juveniles from each of the six jars, placed 4 in individual wells of each of two 6-well multiwell plates, and flipped a coin to assign each plate to a juvenile temperature treatment: either 11°C or 16°C. Then we placed the well plates at the target temperature for a minimum 45 min acclimation period. As such we now had 12 total juveniles (4 from each of three larval replicate jars) in each the following larval→juvenile temperature treatments: (1) 11°C→11°C; (2) 11°C→16°C; (3) 16°C→11°C; (4) 16°C→16°C.

We then randomly assigned the juveniles to 12-chamber bead boxes (two boxes per juvenile temperature treatment, with six juveniles from each relevant treatment in each box), measured them using a dissecting scope on maximum zoom (approximately 50x) fitted with a calibrated ocular micrometer. We added 7 recently-settled (<0.5 mm test diameter) *Strongylocentrotus purpuratus* (purple urchin) juveniles per well as food. We then placed the boxes into the flow-through temperature-controlled juvenile rearing system described above for 20 days, with 3 intervening box cleanings or changes as needed, at which time we recorded numbers of urchins eaten and juvenile size, and replaced the eaten urchins, ensuring that each juvenile was fed ad libitum throughout.

***Exp 3–Settlement 2023* and *Exp 5–Juv Growth 2023*.** In 2023, we repeated *Exp 4* with the following changes (see Table 4 for crosses and fertilization dates; Table 1 for experiment summary). We reared larvae throughout at 3 temperatures, 3 replicate jars each: 11°C, 14°C and 17°C. We also added a fourth treatment of 3 replicate jars at 14°C where we fed the larvae natural plankton only ('14-plankt'). We prepared the natural plankton by collecting ~18 L of raw seawater at the surface off the FHL floating dock (which was approximately 3–6 m above the bottom, depending on the tide). We filtered the raw seawater through a 23 um mesh to remove zooplankton and large protists. We then filtered this filtrate through a 5 μm Nitex mesh, using a vacuum pump, resulting in a 5–23 μm range of plankton being retained on the mesh screen. We then recovered the trapped plankton off the 5 μm filter into MFSW. Based on hemocytometer counts of the recovered plankton cells, we determined that our standard lab diet has approximately 4x more dense plankton than natural seawater. Therefore we fed the collected natural plankton at 4x to the larvae throughout in an attempt to match the amount of food to the lab-grown phytoplankton that the other larvae received. We kept plankton at 8x aerated at 9°C in 12:12 hr light for up to 2 days until use. We note that we fed the natural plankton at 4x throughout, despite observing obvious variation in the collected plankton density on different days.

On 63 dpf, we conducted *Exp 3* by settling the juveniles in 200 ml jars with 2.5 g *C. tuberculosum* and a non-inductive, cultured diatom biofilm (*N. frustulum* and *N. incerta*, see above) rather than a biofilm collected in tanks with sunflower star adults. We then placed 25 larvae from a single replicate into one jar, equilibrated at the larval rearing temperature (therefore 12 total jars, 3 jars per treatment). 11 days later, we measured 12 haphazardly chosen juveniles from each jar under a dissecting scope at maximum zoom (approximately 50x) using a calibrated ocular micrometer, and also recorded how many visible arms the juveniles had.

For *Exp 5–Juv Growth 2023,* we only used juveniles reared at 11 or 17°C, and set up the experiment as in 2022, starting at 11 days post settlement and with the following larval→juvenile temperature treatments: (1) 11°C→11°C; (2) 11°C→17°C; (3) 17°C→11°C; (4) 17°C→17°C. We chose 8 of the 12 juveniles from each jar for the growth experiment based on the post-settlement measurements, excluding the largest and smallest juveniles, as well as any others that showed clear morphological defects. These 8 were then subdivided into 4 nearest-size pairs, with the two juveniles in a pair assigned to one or the other juvenile treatment by coin flip.

We ran the experiment for 6 weeks, feeding the juveniles ad libitum with newly-settled green or purple sea urchin juveniles. We cleaned or changed (as needed) the 12-chamber boxes every 4–6 days, recorded juvenile diameter and other features, as well as number of urchins eaten, and replaced the food.

**Exp 6–Flipping 2023.** We assigned 7−8 month post-settlement juveniles randomly into three treatment groups (based on their experimental test temperature): 12°C (n = 7 juveniles), 14°C (n = 6) and 17°C (n = 8). We acclimated juveniles to the ambient temperature of 12°C for 4 days in individual plexiglass tubes with a 300 µm Nitex mesh screen glued to the bottom, and with the tubes arranged on a perforated platform; submersible water pumps circulated water below, providing water exchange in the tubes. We then conducted the first round of righting behavioral assays ("flipping"; as described below) on each juvenile at 12°C, after which we returned the stars to their sea table and ramped up the temperature the experimental test temperature over the course of a week, in 1.2–1.3°C increments per day, such that the stars were held at the test temperature for a minimum of 48 hours before flipping (see Fig 8).

On the day of the righting assays, starting at noon, we removed juveniles individually from their tubes and placed them in a Petri dish in a water bath held at the experimental temperature. We gently dislodged the juveniles from the Petri dish using a plastic scoopula utensil (see Fig 8) and flipped them 180°, placing them oral side up. We observed the juveniles righting themselves, and then immediately and gently flipped them again. We continuously flipped the juveniles in this manner for 10 minutes while recording the process at 2 frames per second with a USB camera (Basler a2A2600-64ucPRO) controlled by a custom python script. If a juvenile was still righting itself when the 10 minute mark was reached, it was allowed to finish before being returned to the temperature control box. Following flipping we returned the juvenile to its tube and the trial on the next juvenile began. For each star and each flip, we measured righting time (the time between the removal of the scoopula tool by the researcher until the juvenile was again oral side down, with all arms in contact with the Petri dish; see Fig 8) from the video recordings using a custom python graphical user interface.

## Statistics

For *Exp 1*, we modeled the relationship between temperature (Table 2) and each morphological character (S1 Table) using quadratic curves. The temperature corresponding to the curve's peak provides an estimate of the optimal temperature for that feature's development within our experimental conditions. To quantify uncertainty in this estimate, we employed cluster bootstrapping [65]. This involved two steps: sampling replicates with replacement, followed by sampling observations with replacement. We fit a quadratic curve to each bootstrapped sample and identified the temperature at the curve's peak. By repeating this process 1000 times, we generated a 95% confidence interval for the optimal temperature, based on the 2.5% and 97.5% bootstrap quantiles. We used cluster bootstrapping to account for potential 'replicate effects.' This interval reflects the uncertainty in the optimal temperature estimate within the experimental conditions tested.

Because features were highly correlated, we created composite variables to represent larval feature development at the embryo, bipinnaria, and brachiolaria stages, as well as for juvenile feature development at the bipinnaria and brachiolaria stages. This was achieved by performing principal component analysis (PCA) on the respective feature set and selecting the first principal component. The first principal component represents the direction of maximum variance among the features and serves as a composite measure of either larval or juvenile development at the given stage. The variable weightings for these composite variables are provided in S2 Table. We used the same bootstrapping procedure described in the previous paragraph to calculate confidence intervals for the optimal temperature corresponding to this composite measure (see Table 3).

For *Exps 1–5,* we analyzed the data using R (ver. 4.2.3; [66]). We analyzed proportion settled data (Exps 1–2) using a logistic (generalized linear) mixed-effects model, employing the lme4 and emmeans packages [67,68], due to the binomial nature of our response variables (e.g., larvae settled). In our tests, we treated each replicate exposure (jar, etc.) of a group of larvae as a random intercept. In all other experiments from 2022–23 (*Exps 3–5*; see Table 1) we conducted ANOVAs, employing a Tukey HSD test for any post hoc comparisons. These two types of analyses can be distinguished

in Results by the types of statistics shown: we report *Z*-statistics for the logistic mixed-effects models and *F*-statistics for ANOVAs. When we conducted analyses with multiple comparisons, the reported *p*-values are after employing Bonferroni corrections.

For *Exp 6, t*o assess the impact of the temperature treatment (MHW simulation) on righting time, we ran a difference-in-differences (DiD) mixed linear model (reml = True, fit = 'nm'; [69]) from the statsmodels 0.14.4 python library [70] to estimate how righting time changed from the baseline in each experiment. We set righting time as the response variable, with experiment (12->12, 12->14.5, 12->17) and time point in experiment ($T_0$, $T_1$) as categorical variables, with an interaction between experiment and time point in experiment, and individual star ID as a random effect. $T_0$ in each experiment represented the flip tests at the ambient temperature (always 12°C); $T_1$ represented the flip tests at the test temperatures (12, 14.5 or 17°C). For our baseline we used the righting times in the 12->12 control experiment. We conducted a total of n = 601 flips, for N = 21 stars, with a mean count of 28.6 flips per star. We also compared growth rates during the 1 week MHW simulation using a pairwise Welch's T-test with an assumption of unequal variance, followed by Bonferroni-Holm *p*-value correction.

To determine if righting behavior changed over the course of the 10 minute flipping regimen (for example, if the stars became fatigued and righted more slowly as the 10 minutes progressed), we analyzed flip times for every star in all trials as a function of the 10 minute experimental clock time. We saw no evidence for any consistent increase in righting time across any of the 6 experimental runs as a function of time in experiment (3 ambient tests, 3 experimental tests; $t_{600} = -1.307$; $p = 0.19$). In two of the runs –the 12->14 ambient flips ($Z_{5,90} = -1.703$; $p = 0.001$) and the 12->17 test flips ($Z_{7,109} = -2.220$; $p = 0.026$) – there was a significant decrease in righting time as the experiment proceeded.

This latter effect seems to be driven mainly by the first minute of observations, during which there was a high proportion of outlier righting times (see Fig. 9) in some runs when compared to the subsequent 9 minutes. These findings indicate that it may take a short time for some stars to get accustomed to the experimental chamber and the behavioral test. We ran a series of statistical tests to see if these outlier values impacted any of our conclusions, and we found no evidence of such (see Data Availability section for link). We mention it here for researchers proposing to engage in similar experiments, who may wish to give their stars a flipping acclimation period before recording data for analysis.

## Supporting information

**S1 Fig. Embryo stage, larval characters.** Raw larval character measurements in the embryo across different temperature exposures*, Exp 1*. Individual points represent individual larvae, with different colors for each replicate. *Blue lines* show the quadratic fits. *Vertical red lines* show the estimates for optimal temperature for each feature; the *shaded regions* indicate the estimated 95% confidence intervals. See S1 Table for description of characters.
(JPG)

**S2 Fig. Bipinnaria stage, larval characters.** Raw larval character measurements in the bipinnaria (mid) stage larva across different temperature exposures*, Exp 1*. Individual points represent individual larvae, with different colors for each replicate. *Blue lines* show the quadratic fits. *Vertical red lines* show the estimates for optimal temperature for each feature; the *shaded regions* indicate the estimated 95% confidence intervals. See S1 Table for description of characters.
(JPG)

**S3 Fig. Bipinnaria stage, juvenile characters.** Raw juvenile character measurements in the bipinnaria (mid) stage larva across different temperature exposures*, Exp 1*. Individual points represent individual larvae, with different colors for each replicate. *Blue lines* show the quadratic fits. *Vertical red lines* show the estimates for optimal temperature for each feature; the *shaded regions* indicate the estimated 95% confidence intervals. See S1 Table for description of characters and units.
(JPG)

**S4 Fig. Brachiolaria stage, larval characters.** Raw larval character measurements in the brachiolaria (late) stage larva across different temperature exposures, *Exp 1*. Individual points represent individual larvae, with different colors for each replicate. *Blue lines* show the quadratic fits. *Vertical red lines* show the estimates for optimal temperature for each feature; the *shaded regions* indicate the estimated 95% confidence intervals. See S1 Table for description of characters. (JPG)

**S5 Fig. Brachiolaria stage, juvenile characters.** Raw juvenile character measurements in the brachiolaria (late) stage larva across different temperature exposures, *Exp 1*. Individual points represent individual larvae, with different colors for each replicate. *Blue lines* show the quadratic fits. *Vertical red lines* show the estimates for optimal temperature for each feature; the *shaded regions* indicate the estimated 95% confidence intervals. See S1 Table for description of characters and units. (JPG)

**S6 Fig. Juvenile growth during the MHW simulation (*Exp 6*).** (A) Week 1 shows the diameter of each star recorded during the ambient flip test, Week 2 shows the diameter of the same star one week later during the experimental flip test. *Yellow*– stars from 12->12°C treatment; *blue*– 12->14.5°C treatment; *pink*– 12->17°C treatment. (B) Same data as in A but expressed as growth rate: $(Diam_2 - Diam_1)/Diam_1$. (C) Standard box plot of the growth rate data from B. Stars in all three treatments grew measurably ($p < 0.05$). Although we detected no significant differences among treatments in growth rate by standard criteria ($p < 0.05$), there was a hint of faster growth at 12->17°C versus 12->12°C ($p = 0.066$; see the text). (JPG)

**S1 Table. List and description of characters measured or scored in *Exp 1*.** Fig references are either to panels in Fig 1 in this publication, or to panels in Figure 3 in Hodin et al 2021 (*Biol Bull* 241: 243–258). Stage measured: E–embryo; L1–mid larva (bipinnaria); L2–late larva (brachiolaria). (PDF)

**S2 Table.** Weights assigned to individual characters (see S1 Table) for constructing the larval or juvenile PC1 composite variable in *Exp 1*, which models larval or juvenile development at each stage. *Dashes* indicate characters that were not analyzed at that stage. (PDF)

**S3 Table. Larval cloning in *Exp 1* at 64 dpf.** Rates of cloning were higher at 20°C when compared to the other temperature treatments ($Z_{3,15} = 3.134$; $p < 0.01$), though this effect was largely driven by 20°C Replicate C. The four classes of larvae are further described in the Methods. (PDF)

**S4 Table. Larval cloning in *Exp 3* at 63 dpf.** Rates of cloning were 50% higher at 17°C when compared to either 11°C ($Z_{3,9} = 7.518$; $p < 0.001$) or 14°C ($Z_{3,9} = 7.234$; $p < 0.001$). (PDF)

## Acknowledgments

We are grateful to the Friday Harbor Labs staff for unending support. We thank the following people who offered advice or other assistance at key points: Alexi Pearson-Lund, Shannon Cefalu, Pema Kitaeff, Joey Ullmann, Rebecca Guenther, Eric Finn, Merrill Hille, Luke Gardner, Drew Harvell, Sophie George, Richard Strathmann, Aaron Ninokawa, Carla Narvaez, Whiskey Creek Hatchery, Tiffany Rudek, Mark Murray, Ryan Crim, Andreas Heyland, Amro Hamdoun, Jon Allen, Bruno Pernet, Bob Podolsky, Walter Heady, Norah Eddy, Jono Wilson and Mike Nishizaki. We dedicate this work to Tommy Pieples, who was always there when we needed him.

## Author contributions

**Conceptualization:** Jason Hodin, Fleur P. Anteau.

**Data curation:** Jason Hodin, Fleur P. Anteau, Michael Brito, Fiona Curliss, Chloe J. Schwab, Vannessa V. Valdez, Willem Weertman.

**Formal analysis:** Jason Hodin, James S. Peng, Willem Weertman.

**Funding acquisition:** Jason Hodin.

**Investigation:** Jason Hodin, Fleur P. Anteau, Brook F. Ashcraft, Michael Brito, Fiona Curliss, Augustin R. Kalytiak-Davis, Willem Weertman.

**Methodology:** Jason Hodin, Fleur P. Anteau, Brook F. Ashcraft, Michael Brito, Fiona Curliss, Augustin R. Kalytiak-Davis, Chloe J. Schwab, Vannessa V. Valdez, Willem Weertman.

**Project administration:** Jason Hodin.

**Software:** Willem Weertman.

**Supervision:** Jason Hodin.

**Validation:** Jason Hodin, Willem Weertman.

**Visualization:** Jason Hodin, Willem Weertman.

**Writing – original draft:** Jason Hodin.

**Writing – review & editing:** Jason Hodin, Fleur P. Anteau, Brook F. Ashcraft, Michael Brito, Fiona Curliss, Augustin R. Kalytiak-Davis, James S. Peng, Chloe J. Schwab, Vannessa V. Valdez, Willem Weertman.

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
