## [Decision Letter · Decision Letter 0]

28 Apr 2025

PONE-D-25-03720Star Power: Early life stages of an endangered sea star are robust to current and near-future warmingPLOS ONE

Dear Dr. Hodin,

Thank you for submitting your manuscript to PLOS ONE. After careful consideration, we feel that it has merit but does not fully meet PLOS ONE’s publication criteria as it currently stands. Therefore, we invite you to submit a revised version of the manuscript that addresses the points raised during the review process.

**The reviewer raised two minor questions that should be addressed in the manuscript if possible.  Note that addressing these concerns by executing respective experiments is optional (as it might be beyond the scope of this manuscript).  **

We look forward to receiving your revised manuscript.

Kind regards,

Sebastian D. Fugmann, Ph.D.

Academic Editor

PLOS ONE

**Journal Requirements:**

1. When submitting your revision, we need you to address these additional requirements. Please ensure that your manuscript meets PLOS ONE's style requirements, including those for file naming. The PLOS ONE style templates can be found at https://journals.plos.org/plosone/s/file?id=wjVg/PLOSOne_formatting_sample_main_body.pdf and https://journals.plos.org/plosone/s/file?id=ba62/PLOSOne_formatting_sample_title_authors_affiliations.pdf 2. Please update your submission to use the PLOS LaTeX template. The template and more information on our requirements for LaTeX submissions can be found at http://journals.plos.org/plosone/s/latex. 3. Thank you for stating the following in the Acknowledgments Section of your manuscript: We are grateful to the Friday Harbor Labs staff for unending support. We thank the following people who offered advice or other assistance at key points: Alexi Pearson-Lund, Shannon Cefalu, Pema Kitaeff, Joey Ullmann, Rebecca Guenther, Eric Finn, Merrill Hille, Luke Gardner, Drew Harvell, Sophie George, Richard Strathmann, Aaron Ninokawa, Carla Narvaez, Whiskey Creek Hatchery, Tiffany Rudek, Ryan Crim, Andreas Heyland, Amro Hamdoun, Jon Allen, Bruno Pernet, Bob Podolsky, Walter Heady, Norah Eddy, Jono Wilson and Mike Nishizaki. The Nature Conservancy of California, the Ocean Protection Council and California SeaGrant provided financial support, as did dozens of donors. We dedicate this work to Tommy Pieples, who was always there when we needed him. We note that you have provided funding information that is not currently declared in your Funding Statement. However, funding information should not appear in the Acknowledgments section or other areas of your manuscript. We will only publish funding information present in the Funding Statement section of the online submission form. Please remove any funding-related text from the manuscript and let us know how you would like to update your Funding Statement. Currently, your Funding Statement reads as follows: JH – The Nature Conservancy of California (TNC-CA; no funding number), participated in conversations about some aspects of study design; TNC-CA did not participate in any data collection, analyses or manuscript preparation.JH – California Sea Grant (CSG) and the Ocean Protection Council (CSG award number R/HCE-15), did not participate in any aspects of study design, data collection, data analysis or manuscript preparation  Please include your amended statements within your cover letter; we will change the online submission form on your behalf. 4. In the online submission form, you indicated that your data will be submitted to a repository upon acceptance.  We strongly recommend all authors deposit their data before acceptance, as the process can be lengthy and hold up publication timelines. Please note that, though access restrictions are acceptable now, your entire minimal dataset will need to be made freely accessible if your manuscript is accepted for publication. This policy applies to all data except where public deposition would breach compliance with the protocol approved by your research ethics board. If you are unable to adhere to our open data policy, please kindly revise your statement to explain your reasoning and we will seek the editor's input on an exemption. 5. When completing the data availability statement of the submission form, you indicated that you will make your data available on acceptance. We strongly recommend all authors decide on a data sharing plan before acceptance, as the process can be lengthy and hold up publication timelines. Please note that, though access restrictions are acceptable now, your entire data will need to be made freely accessible if your manuscript is accepted for publication. This policy applies to all data except where public deposition would breach compliance with the protocol approved by your research ethics board. If you are unable to adhere to our open data policy, please kindly revise your statement to explain your reasoning and we will seek the editor's input on an exemption. Please be assured that, once you have provided your new statement, the assessment of your exemption will not hold up the peer review process. 6. We note that you have included the phrase “data not shown” in your manuscript. Unfortunately, this does not meet our data sharing requirements. PLOS does not permit references to inaccessible data. We require that authors provide all relevant data within the paper, Supporting Information files, or in an acceptable, public repository. Please add a citation to support this phrase or upload the data that corresponds with these findings to a stable repository (such as Figshare or Dryad) and provide and URLs, DOIs, or accession numbers that may be used to access these data. Or, if the data are not a core part of the research being presented in your study, we ask that you remove the phrase that refers to these data. 7. We note that you have referenced “COSEWIC” which has currently not yet been accepted for publication. Please respond by return e-mail with a copy of your updated manuscript to include to remove this from your References and amend this to state in the body of your manuscript: (COSEWIC. [in preparation]”) as detailed online in our guide for authorshttp://journals.plos.org/plosone/s/submission-guidelines#loc-reference-style.   We can then upload this to your submission on your behalf. 8. Please review your reference list to ensure that it is complete and correct. If you have cited papers that have been retracted, please include the rationale for doing so in the manuscript text, or remove these references and replace them with relevant current references. Any changes to the reference list should be mentioned in the rebuttal letter that accompanies your revised manuscript. If you need to cite a retracted article, indicate the article’s retracted status in the References list and also include a citation and full reference for the retraction notice.

Reviewers' comments:

Reviewer's Responses to Questions

**Comments to the Author**

1. Is the manuscript technically sound, and do the data support the conclusions?

Reviewer #1: Yes

2. Has the statistical analysis been performed appropriately and rigorously? 

Reviewer #1: Yes

3. Have the authors made all data underlying the findings in their manuscript fully available?

Reviewer #1: Yes

4. Is the manuscript presented in an intelligible fashion and written in standard English?

Reviewer #1: Yes

5. Review Comments to the Author

**Reviewer #1:**  The authors present a series of experiments aimed at determining the effects of temperature on different stages in the life history of the sunflower sea star, Pycnopodia helianthoides. The abundance of this sea star has been severely reduced in its southern range by sea star wasting disease and this has implications for the wider ecosystem. Populations remain healthy in its northern range (e.g., off Washington state/British Columbia in the Salish sea) and the authors assess the tolerance of several life history stages to higher temperatures that may be encountered in more southern pacific coast habitats and as ocean temperatures rise in general. The overarching goal is to assess whether animals from Salish sea populations would be suitable to repopulate decimated southern populations.

To get at this problem the authors set up several comparisons where growth and maturation are measured across different temperatures using embryos and larvae, settlement-competent late larvae, and post-settlement juveniles. Five experiments measure different developmental aspects with respect to temperature: (1) embryo/larval growth, (2) the effects pre-settlement shifts, (3) settlement, (4) juvenile growth, and (5) juvenile righting performance as a measure of juvenile health. Cloning rates are also measured as an incidental observation.

The experiments are described in detail, the methods are clearly outlined, and issues arising during the investigation are addressed. the findings are sometimes unexpected (e.g., a higher temperature optimum for larval growth and temperature effects on cloning). Findings are presented with cautious interpretation and detailed explanations. The findings are relevant to their original goal of assessing whether northern sunflower sea star populations may be useful in the repopulation of southern habitats.

Overall, the research represents a meaningful basis for investigating sea temperature changes across the early life stages of the sunflower sea star. This has both practical importance in conservation and repopulation programs and interesting biological importance in the differential effects on larval and developing juvenile growth. The detailed descriptions of methods and problems encountered in the study also serve as a model for similar studies on other marine larvae. I have only one point that may be worth considering below.

Specific point: Since the system under consideration consists of both larvae and their food algae, are temperature effects on the algae also something that may affect growth? Or temperature effects of settlement substrate diatoms, coralline algae?

6. PLOS authors have the option to publish the peer review history of their article (what does this mean? ). If published, this will include your full peer review and any attached files.

**Do you want your identity to be public for this peer review?** For information about this choice, including consent withdrawal, please see our Privacy Policy .

Reviewer #1: **Yes: ** Jonathan P. Rast

---

## [Editor Report · Decision Letter 1]

18 Jun 2025

Star Power: Early life stages of an endangered sea star are robust to current and near-future warming

PONE-D-25-03720R1

Dear Dr. Hodin,

We’re pleased to inform you that your manuscript has been judged scientifically suitable for publication and will be formally accepted for publication once it meets all outstanding technical requirements.

Kind regards,

Sebastian D. Fugmann, Ph.D.

Academic Editor

PLOS ONE
---

## [Editor Report · Acceptance letter]

PONE-D-25-03720R1

PLOS ONE

Dear Dr. Hodin,

I'm pleased to inform you that your manuscript has been deemed suitable for publication in PLOS ONE. Congratulations! Your manuscript is now being handed over to our production team.

Kind regards,

on behalf of

Dr. Sebastian D. Fugmann

Academic Editor

PLOS ONE